# THE-TREE: CAN TRACING HISTORICAL EVOLUTION ENHANCE SCIENTIFIC VERIFICATION AND REASONING?

## ABSTRACT

Large Language Models (LLMs) are accelerating scientific idea generation, but rigorously evaluating these numerous, often superficial, AI-generated propositions for novelty and factual accuracy is a critical bottleneck; manual verification is too slow. Existing validation methods are inadequate: LLMs as standalone verifiers may hallucinate and lack domain knowledge (our findings show 60% unawareness of relevant papers in specific domains), while traditional citation networks lack explicit causality and narrative surveys are unstructured, underscoring the absence of structured, verifiable, and causally-linked historical data of scientific evolution. To address this, we introduce **THE-Tree** (**T**echnology **H**istory **E**volution **Tree**), a computational framework that constructs such domain-specific evolution trees from scientific literature. THE-Tree employs a search algorithm to explore evolutionary paths using a novel "Think-Verbalize-Cite-Verify" process: an LLM proposes potential advancements and cites supporting literature, while each proposed evolutionary link is validated for logical coherence and evidential support by interrogating the cited literature. We construct and validate 88 THE-Trees across diverse domains and release a benchmark dataset including up to 71k fact verifications covering 27k papers to foster further research. Experiments demonstrate that i) in graph completion, our THE-Tree improves hit@1 by 8% to 14% across multiple models compared to traditional citation networks; ii) for predicting future scientific developments, it improves hit@1 metric by nearly 10%; and iii) when combined with other methods, it boosts the performance of evaluating important scientific papers by almost 100%. By constructing explicit, verifiable pathways of scientific progression, THE-Tree provides a robust historical foundation for evaluating new hypotheses (human or AI-generated) and enables a computable science history, fostering evidence-based AI-driven scientific discovery.

## 1 INTRODUCTION

Automating scientific discovery has been a long-standing goal (Langley, 1987; Hutter, 2000). The recent rise of Large Language Models (LLMs) offers new avenues, with applications from hypothesis generation (Yang et al., 2023; Boiko et al., 2023a; Baek et al., 2024) to simulating autonomous AI scientists (Bran et al., 2023; Boiko et al., 2023b). However, a critical bottleneck remains: the effective evaluation and validation of scientific ideas, whether AI or human-generated.

Current idea validation approaches face several critical challenges. First, manual verification, while ideal, is prohibitively time-consuming (Si et al., 2024). Second, automated validation using LLMs (Huang et al., 2023; Baek et al., 2024) exhibits multiple limitations: (1) potential for hallucination and incomplete domain knowledge (our findings show 60% unawareness of relevant papers in specific domains), (2) susceptibility to superficial textual features, often highly rating plausible but erroneous propositions (Si et al., 2024; Yang et al., 2023), and (3) inheritance of biases when trained on human review data (e.g., CycleReviewer (Weng et al., 2025)). Third, existing knowledge representation methods are inadequate, as (1) citation networks (Bornmann & Daniel, 2008; Hao et al., 2024) contain noise and lack explicit causal links, and (2) narrative surveys remain unstructured. These limitations fundamentally stem from the absence of structured, causally linked historical data, hindering reliable

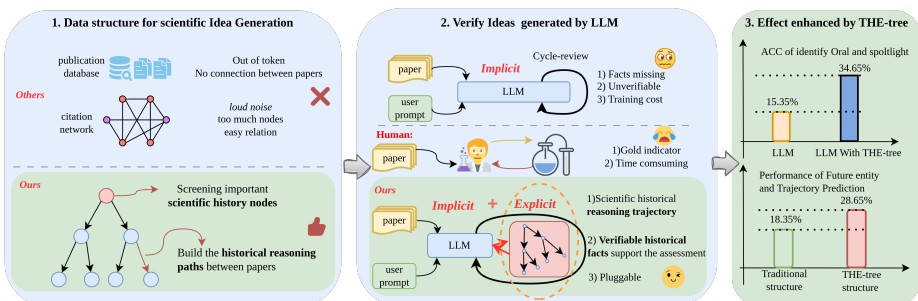

Figure 1: **Overview of THE-Tree.** (1) Limitations of existing data structures (publication databases, citation networks) for scientific idea generation versus THE-Tree's approach of constructing historical reasoning connections by screening important scientific history nodes and building pathways between them. (2) Methods for verifying LLM-generated ideas: implicit LLM-based cycle-review, human evaluation, and explicit THE-Tree-enhanced LLM verification. (3) Comparison of verification methods, highlighting issues like fact missing in LLM-only approaches, high cost in human evaluation, and THE-Tree's use of verifiable historical facts and scientific reasoning trajectories. (4) Performance improvements with THE-Tree in different tasks

AI-driven idea validation. These challenges stem from a critical absence of structured, causally linked historical data, hindering reliable AI-driven validation of ideas.

To address this, we propose leveraging the authentic patterns and causal evolutionary pathways from scientific history for more reliable assessment. We introduce **THE-Tree** (**T**echnology **H**istory **E**volution **Tree**), a computational framework to construct structured, verifiable, domain-specific technology evolution trees from scientific literature (illustrated in Figure 1). THE-Tree builds a topic's evolution by representing individual papers as nodes and the inferential relationships between these papers, specific to the topic, as edges. This aims to provide a solid factual basis and clear historical context for evaluating new hypotheses.

THE-Tree utilizes a Self-Guided Temporal Monte Carlo Tree Search (SGT-MCTS) and a novel Think-Verbalize-Cite-Verify (TVCV) methodology for node expansion. This process prompts an LLM to generate potential evolutionary steps (Think), summarize them concisely (Verbalize), ground them in specific supporting literature (Cite), and critically, validate the proposed relationship (Verify). Crucially, the 'Verify' step employs a Retrieval-Augmented Natural Language Inference (RA-NLI) mechanism to assess the causal and logical coherence of proposed relationships based on cited evidence, ensuring semantic soundness and fidelity of the identified evolutionary relationships. Tree construction often starts from human-validated knowledge like scientific surveys, providing a reliable starting point.

We demonstrate THE-Tree's efficacy by constructing trees for 88 distinct topics across relevant scientific domains. Their quality, validated by automated metrics and human assessment, confirms their effectiveness in reconstructing meaningful and accurate technological trajectories. Downstream tasks, including future node prediction and graph completion, showcase THE-Tree's potential to enhance AI-assisted scientific reasoning by anticipating subsequent developments and highlight its superiority over traditional citation networks. For instance, in graph completion, it outperforms traditional citation networks (e.g., on hit@1 across all tested models); for future node and trajectory prediction, it improves hit@1 by nearly 10%; and in paper evaluation, it enhances the ability of other models to assess important papers by almost 100%. In summary, our main contributions are:

- A novel computational framework, THE-Tree, incorporating the TVCV methodology with LLM-guided SGT-MCTS to construct and validate verifiable technology evolution trees from scientific literature, addressing the lack of structured, causal historical data for scientific evaluation.

- A RA-NLI mechanism within TVCV for rigorous validation of the logical and causal coherence of evolutionary relationships, ensuring tree fidelity.

- The construction and validation of THE-Trees dataset, comprising 88 technology evolution trees across AI domains and a benchmark dataset of 71k fact verification evaluations from 27k scientific papers, with extensive experiments demonstrating superior performance in downstream tasks.

- A structured, verifiable foundation based on historical evolution patterns for evaluating scientific ideas (both human and AI-generated) and supporting grounded AI-driven discovery processes, enabling systematic assessment of scientific progress.

## 2 RELATED WORK

Our work intersects with several research areas, primarily AI for scientific discovery, scholarly knowledge representation, and the evaluation of scientific novelty.

**AI for Scientific Discovery and Evaluation.** The ambition to automate scientific discovery using AI has gained significant momentum with the rise of LLMs (Hutter, 2001; Jansen et al., 2025). Current applications range from hypothesis generation (Yang et al., 2023; Boiko et al., 2023a; Baek et al., 2024) to simulating autonomous AI agents for specific scientific tasks (Bran et al., 2023; Boiko et al., 2023b). A persistent challenge, as highlighted in our Introduction, is the rigorous evaluation of the novelty and feasibility of AI-generated outputs. Manual verification remains a bottleneck (Ferdowsi et al., 2024). While some emerging efforts aim to automate aspects of this evaluation, they often focus on simulating existing human processes (Huang et al., 2023; Lu et al., 2024; Weng et al., 2025). For instance, CycleResearcher (Weng et al., 2025) employs an LLM within an automated research-review loop to mimic peer review by predicting scores and providing feedback. Although valuable for replicating current assessment workflows, such approaches primarily model established paradigms and may not provide the deep. Our work complements these efforts by focusing on constructing the underlying historical structure itself, offering a fact-based pathway for evaluation grounded in demonstrable scientific lineage.

**Scholarly Knowledge Representation and Analysis.** Representing and analyzing the vast body of scientific literature has long been a goal. Traditional bibliometric methods, including citation analysis (Garfield, 1979; Small, 1973) and co-word analysis (Callon et al., 1983), offer insights into publication impact and thematic trends. Science mapping tools like VOSviewer (Van Eck & Waltman, 2010) and CiteSpace (Chen, 2006) provide valuable visualizations of research landscapes. However, as noted in our Introduction, these approaches face limitations. Citation networks are often noisy and fail to capture the explicit causal or logical dependencies signifying true intellectual inheritance (Bornmann & Daniel, 2008), making them a "poor substrate for tracing idea lineage". Co-word analysis identifies term co-occurrence but not necessarily causal links. While useful for broad overviews, these methods generally lack the granularity and causal structure needed for deep reasoning about technological evolution or predictive analysis of research trajectories.

**Scholarly Knowledge Graphs.** More recently, large-scale scholarly knowledge graphs, such as the Microsoft Academic Knowledge Graph (Wang et al., 2020) and AMiner (Tang et al., 2008), have emerged, integrating diverse metadata. Knowledge graph construction techniques (Auer & Stocker, 2018) have also been applied to scientific literature. While these graphs offer rich resources, they often focus on entity relationships (e.g., author collaborations, affiliations) or represent relatively static snapshots of knowledge domains. They typically do not explicitly model the dynamic, temporal, and causal evolutionary pathways of scientific ideas – how one concept or technology directly enables or influences the next. Capturing this validated, directed evolution is precisely the gap THE-Tree aims to fill.

## 3 THE-TREE AS SCIENTIFIC VERIFIER

### 3.1 THE-TREE: A STRUCTURED REPRESENTATION OF SCIENTIFIC EVOLUTION

The pursuit of scientific discovery is increasingly aided by AI, yet verifying the novelty and validity of numerous AI-generated or human-conceived hypotheses presents a significant bottleneck. Traditional methods like citation networks lack semantic depth, offering only noisy and superficial links, while LLMs as standalone verifiers can hallucinate or miss crucial domain knowledge. This underscores the urgent need for a structured, verifiable, and causally-linked representation of scientific evolution.

To address this challenge, we introduce the **THE-Tree (Technology History Evolution Tree)**. In a THE-Tree, each scientific paper is conceptualized as a **node** containing rich metadata including title, abstract, authors, publication venue, and year. Each node $v$ is associated with an importance score $S_v$ reflecting its relevance and impact within the specific domain (detailed in Section 3.3.1).

**Formal Edge Definition:** Let $e_{i,j} = (v_i, v_j, r_{i,j}, \tau_{i,j})$ represent a directed edge from paper node $v_i$ to $v_j$, where $r_{i,j} \in \{$causal, enabling, foundational$\}$ denotes the relationship type, and $\tau_{i,j} \in [0, 1]$ represents the confidence score derived from RA-NLI validation. An edge exists if and only if: (1) $Y_{v_i} \leq Y_{v_j}$ (temporal consistency), (2) $\tau_{i,j} \geq \theta_{min}$ where $\theta_{min} = 0.7$ (validation threshold), and (3) there exists verifiable textual evidence $E_{i,j}$ from $v_j$ that explicitly acknowledges the contribution of $v_i$ to the ideas presented in $v_j$. The edges represent the *historical, inferential, and evolutionary relationships* between these paper nodes (see Figure 3). Unlike traditional citation networks that provide only superficial "cite" relationships, THE-Tree edges capture deeper semantic meanings—how papers causally contribute to, enable, or provide foundations for subsequent advances.

**Handling Paradigm Shifts and Revolutionary Advances:** THE-Tree is designed to capture both incremental and revolutionary scientific advances. For paradigm shifts, our framework employs specialized heuristics: (1) identifying foundational papers that initiate new research directions through high citation bursts and novel terminology introduction, (2) detecting conceptual discontinuities where new approaches fundamentally challenge existing assumptions, and (3) modeling parallel evolutionary branches that may eventually converge or compete. Revolutionary advances often exhibit temporal gaps in the evolution tree, requiring our TVCV methodology to validate conceptual leaps through rigorous evidence retrieval. This enables THE-Tree to represent both continuous technological progression and disruptive innovations that reshape entire research landscapes. Our methodology (detailed in Section 3.3.1) focuses on establishing edges that reflect substantive intellectual lineage, filtering out noise to capture true innovation pathways.

The primary purpose of constructing and utilizing THE-Trees is to provide a *structured, verifiable, and causally-linked historical tapestry of scientific evolution* for specific domains. This graph-based representation serves as a robust knowledge scaffold, allowing new scientific propositions to be situated within explicit, evidence-backed evolutionary contexts. By tracing connections and analyzing pathways within THE-Tree, we can rigorously assess a new idea's novelty, its factual consistency with established knowledge, and its potential impact, thereby addressing the limitations of standalone LLM evaluators. THE-Tree thus facilitates a more evidence-based approach to scientific idea validation.

## 3.2 Leveraging THE-Tree for Scientific Idea Verification

Once a THE-Tree, with its richly annotated nodes and semantically meaningful edges, is constructed for a specific scientific domain, it serves as a powerful instrument for the verification and contextualization of new scientific ideas or papers. We propose a straightforward yet effective methodology to utilize THE-Tree for this purpose, enabling the retrospection of relevant historical evolutionary paths for a given input scientific paper, $P_{in}$, defined by its title $T_{in}$ and abstract $A_{in}$. This approach provides critical context by situating new research within established knowledge frameworks, thereby aiding in the assessment of its novelty and potential contribution.

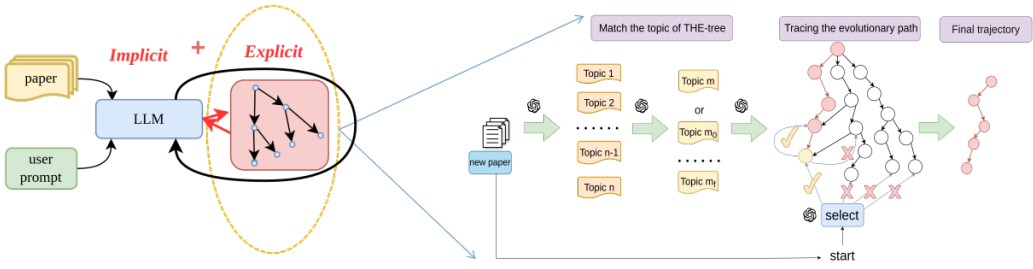

Figure 2: **THE-Tree utilization for scientific idea verification.** Illustration of the methodology for leveraging THE-Tree to verify new scientific ideas through historical evolutionary path retrospection, showing the process from input paper analysis to relevant pathway identification.

The core steps of this verification and retrospection process are as follows:

1. **Initialization**: This methodology assumes access to: a) A collection of pre-computed THE-Trees $\{G_k\}$, where each $G_k = (V_k, E_k)$ corresponds to a specific scientific topic $Topic_k$. Nodes $v \in V_k$ represent scientific papers with attributes such as publication year $Y_v$, title $T_v$, abstract $A_v$, and an importance score $S_v$, as defined in Section 3.1. Edges $e \in E_k$ signify directed evolutionary relationships, capturing inferential and developmental dependencies. b) A Large Language Model (LLM) for semantic tasks such as similarity assessment and topic matching.

2. **Topic Space Identification**: For the input paper $P_{in}$, a set of $N_T$ most relevant topics, $\mathcal{T}_{P_{in}} = \{Topic_1, \ldots, Topic_{N_T}\}$, is identified from the available THE-Tree topics. This involves prompting an LLM with $T_{in}$, $A_{in}$, and the list of all THE-Tree topic names to determine the most appropriate THE-Tree(s) for situating $P_{in}$.

3. **Candidate Path Origination**: For each relevant THE-Tree $G_k$: a) Identify recent papers $\{v_{k,j}\} \subset V_k$ based on publication year, b) Calculate semantic similarity scores $Sim(P_{in}, v_{k,j})$ between $P_{in}$ and candidates using LLM techniques, c) Select top $N_S$ papers $\mathcal{S}_k = \{v^*_{k,1}, \ldots, v^*_{k,N_S}\}$ with similarity $\geq \theta_{sim}$ as terminal nodes for backward path tracing.

4. **Historical Path Retrospection**: For each terminal paper $v^*_{term}$, construct historical path $Path_{k,l} = (v_1, v_2, \ldots, v_n = v^*_{term})$ by iteratively selecting predecessors using lexicographical optimization:

$$v^*_{pred} = \arg \min_{v \in Pred(v_{curr}), Y_v \leq Y_{v_{curr}}} ((Y_{v_{curr}} - Y_v), -S_v)$$

   This prioritizes predecessors with smallest time gaps and highest importance scores. Traversal continues until no valid predecessor is found or maximum path length is reached.

5. **Path Aggregation and Presentation**: Collect all paths $\{Path_{k,l}\}$ and rank them based on similarity scores, cumulative node importance, or path coherence. Select top $N_P$ paths for presentation with paper sequences, key attributes, and evolutionary relationships.

This methodology provides a simple yet powerful way to leverage the structured knowledge within THE-Trees to verify a new scientific idea by exploring its historical context and connections to established research. The retrieved paths can highlight the foundations upon which $P_{in}$ builds, identify potentially overlooked prior art, or help assess its incremental novelty versus a more radical departure from existing trajectories. Users can adapt this general approach based on specific analytical needs, such as modifying the LLM prompts, similarity thresholds, path selection heuristics, or the depth of retrospection.

## 3.3 AUTOMATED CONSTRUCTION OF THE-TREE

The manual construction of comprehensive and accurate THE-Trees for diverse scientific domains would be a prohibitively laborious task. Therefore, we develop a computational framework for the automated construction of THE-Trees from scientific literature. Our approach formulates this construction as an optimization problem: the goal is to identify and assemble evolutionary paths through the literature that maximize a composite reward. This reward is designed to reflect the significance of the constituent papers (nodes) and the logical coherence and evidential support of the evolutionary steps (edges) they represent:

$$\max_{\text{Path}} \left( \sum_{v \in \text{Path}} S(v) + R_{\text{gen}} + R_{\text{attr}} \right), \tag{1}$$

where $S(v)$ is the importance score of a node (paper) $v$ (see Section 3.3.1 for how $S(v)$ is determined), $R_{\text{gen}}$ is the generation process reward reflecting the coherence of the path (how well a new node continues an existing path), and $R_{\text{attr}}$ is the attribution process reward validating the evidential support for the link (edge) between connected nodes. This objective is pursued using a Self-Guided Temporal Monte Carlo Tree Search (SGT-MCTS) algorithm, as detailed in the pipeline stages below.

### 3.3.1 THE-TREE CONSTRUCTION PIPELINE OVERVIEW

The construction of a THE-Tree involves several key stages, from initial data preparation to the iterative refinement of the tree structure. Figure 3 provides a schematic overview of this pipeline.

Figure 3: **THE-Tree construction overview.** (a) Extracting chunks and references from surveys to build an initial concept graph structure; (b) Generating a structured knowledge tree using the SGT-MCTS algorithm, guided by the TVCV methodology for node expansion and RA-NLI for relationship validation.

**Dataset Construction: Initialization and Foundation.** THE-Tree construction begins with scientific surveys that encapsulate rich historical information about scientific progression. We transform these unstructured narratives into structured datasets by strategically selecting surveys from diverse time periods to mitigate recency bias. The output comprises interconnected survey documents, paper nodes (from citations), concept nodes (extracted ideas), and their explicit relationships, overcoming limitations of raw narratives and noisy citation networks (details in Appendix A.8).

**Core Component: Self-Guided Temporal Monte Carlo Tree Search (SGT-MCTS).** We employ Self-Guided Temporal Monte Carlo Tree Search (SGT-MCTS) (Browne et al., 2012) to navigate potential technological evolutionary paths. SGT-MCTS iteratively builds the tree by selecting nodes for expansion, simulating paths, and backpropagating rewards (Algorithm 1), guided by the composite reward function (Equation 1), balancing node importance, path coherence, and link validity. The key components include: **a) LLM-Enhanced Node Importance Assessment ($S(v)$):** The importance of a paper (node $v$) is a weighted combination of its structural significance within the citation graph and its semantic relevance as assessed by an LLM: $S(v) = \gamma \cdot S_{\text{graph}}(v) + (1 - \gamma) \cdot S_{\text{LLM}}(v)$, where $\gamma$ is a weighting factor. $S_{\text{graph}}(v)$ combines multiple centrality measures (PageRank, citation count, degree, betweenness, and eigenvector centrality) with dynamic weighting. $S_{\text{LLM}}(v)$ is obtained by prompting an LLM to assess the paper's importance within the specified topic. **b) Generation Process Reward ($R_{\text{gen}}$):** This reward encourages the formation of coherent and temporally sound evolutionary paths. It is defined for a node $v$ given a preceding path $P_{prev}$: $R_{\text{gen}}(v|P_{prev}) = \text{DPO}(v|P_{prev}) \cdot \text{TemporalCoherence}(v|P_{prev})$. The DPO score (Rafailov et al., 2023), approximated by an LLM, evaluates how well node $v$ continues the trajectory of $P_{prev}$. The Temporal Coherence term penalizes achronological or large time gaps. Node selection uses our SGT-UCT variant incorporating LLM guidance and temporal coherence (Kocsis & Szepesvári, 2006):

$$\text{SGT-UCT}(v) = \left( \frac{Q(v)}{N(v)} + c \cdot \sqrt{\frac{\ln N(p)}{N(v)}} + \lambda \cdot \text{LLM}_{\text{priority}}(v) \right) \cdot \text{TempCoherence}(v|P_{prev}) \quad (2)$$

This balances exploitation $Q(v)/N(v)$, exploration $c \cdot \sqrt{\ln N(p)/N(v)}$, LLM guidance $\text{LLM}_{\text{priority}}(v)$, and temporal consistency $\text{TempCoherence}(v|P_{prev})$.

**Node Expansion Method: Think-Verbalize-Cite-Verify (TVCV) Methodology.** Node expansion within SGT-MCTS, the process of adding new paper nodes and establishing connections (edges with rich semantic meaning as discussed in Section 3.1), is performed using our novel Think-Verbalize-Cite-Verify (TVCV) methodology (see Algorithm 2). This process leverages an LLM to systematically generate, ground, and validate new nodes (potential technological advancements) and their links within the tree:

- **Think:** The LLM generates candidate technological advancements or scientific contributions that could logically follow from the current path history and domain knowledge.

- **Verbalize:** The LLM summarizes these generated ideas into concise statements or propositions that represent potential new nodes.

- **Cite:** For each summarized proposition, the LLM retrieves or identifies specific supporting scientific literature (i.e., existing paper nodes from the dataset or newly found papers) that grounds the proposed advancement, thereby proposing a potential link between an existing paper node and a new one.

- **Verify:** The proposed relationship (edge) between the current path's terminal node and the newly cited paper node, along with the relevance of the new node itself, is rigorously validated for logical consistency, causal coherence, and temporal soundness. This crucial step ensures the factual and logical soundness of the link, and is performed by the RA-NLI mechanism described in Section 3.3.1.

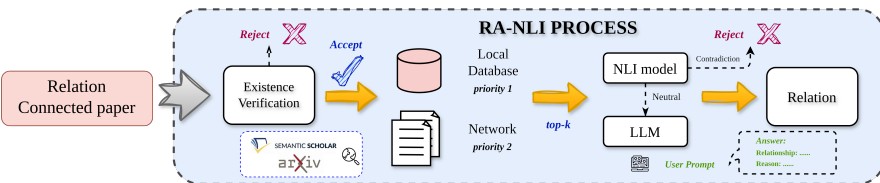

Figure 4: **Overview of RA-NLI process.** This figure illustrates the RA-NLI process, including citation existence verification, document retrieval, and semantic relation assessment using NLI and LLM, which forms the core of the *Verify* step in TVCV and the $R_{\text{attr}}$ calculation.

**Relationship Validation: Retrieval-Augmented Natural Language Inference (RA-NLI) Mechanism.**

- **Addressing LLM Hallucination:** Our TVCV methodology with RA-NLI verification eliminates phantom citations entirely (from 21.46% to 0%) compared to simplified TVC approaches, significantly reducing hallucination rates (detailed analysis in Appendix A.4).

- **Expert Refinement:** Domain experts validate and refine automatically generated pathways, achieving 15.3% average coherence improvements (detailed protocol in Appendix A.2).

  The critical *Verify* step of the TVCV methodology (Section 3.3.1), and the basis for the attribution process reward ($R_{\text{attr}}$) in Equation 1, is performed by our Retrieval-Augmented Natural Language Inference (RA-NLI) mechanism. RA-NLI is designed to rigorously assess the causal and logical coherence of the evolutionary link (edge) proposed between a parent node (e.g., $v_{\text{parent}}$) and a newly cited child node ($v$),

- **Computational Considerations:** THE-Tree construction requires significant computational resources. Using Qwen2.5-72B, a typical tree takes approximately 4.73 hours on 8×A100 GPUs, achieving Entity F1=0.75 and Relation F1=0.70. Smaller models offer faster generation (7B models: 2.42 hours, 49% faster) but with reduced accuracy (Entity F1=0.65), necessitating more human correction effort. Importantly, THE-Tree is designed as a one-time build creating reusable, long-term knowledge foundations that amortize the initial computational investment across multiple downstream applications.

$$R_{\text{attr}}(v_{\text{parent}} \rightarrow v) = \text{RA-NLI}(s_{\text{parent}}, s_v) \tag{3}$$

- **RA-NLI Technical Overview:** Our RA-NLI mechanism operates through a three-stage pipeline of retrieval, inference, and aggregation with confidence thresholding (detailed technical parameters in Appendix A.3). This mechanism (illustrated in Figure 4) integrates embedding-based retrieval to fetch relevant contextual passages from the cited documents, followed by fine-grained Natural Language Inference using a specialized model ($f_{\text{NLI}}$) to determine if the textual description of the child node ($s_v$) is entailed, contradicted by, or neutral with respect to the parent node's description ($s_{\text{parent}}$) and supporting evidence. An LLM-based verifier ($f_{\text{LLM}}$) further refines ambiguous cases. This ensures semantic soundness and high fidelity for the identified evolutionary relationships, forming the backbone of the THE-Tree's verifiability.

## 4 EXPERIMENTS AND RESULTS

### 4.1 EXPERIMENTAL VALIDATION STRATEGY

Our experiments directly address the core claims of THE-Tree's effectiveness. We validate three key hypotheses: (1) *Quality Reconstruction* (Section 4.2): THE-Tree can accurately reconstruct meaningful technology trajectories, validated through expert comparison and automated metrics showing 75% entity F1 and 70% relation F1; (2) *Superior Knowledge Representation* (Section 4.3): THE-Tree captures richer structural information than citation networks, demonstrated through graph completion tasks with consistent 10-15% improvements across Hit@k metrics; (3) *Enhanced Scientific Evaluation* (Section 4.1): THE-Tree augmentation significantly improves LLM-based paper assessment, nearly doubling accuracy in identifying high-impact papers. This systematic validation directly supports our motivation that structured evolutionary history enables more reliable scientific verification.

For detailed information on dataset construction, statistics, and the definitions of evaluation metrics, please refer to Appendix B.1 and Appendix B.3 respectively.

**Algorithm 1** SGT-MCTS Implementation

**Input:** Root node representing the starting technology concept
**Output:** Comprehensive technology evolution tree
1: Initialize the tree with the root node
2: **while** not converged **do**
3:    **Select:** Choose promising node
4:    node ← select(root)     ▷ SGT-UCT scoring (Eq. 2)
5:    **Expand:** Generate new technology nodes
6:    new_nodes ← expand(node)    ▷ TVCV methodology (Alg. 2, Sec. 3.3.1)
7:    **Simulate:** Evaluate potential paths
8:    reward ← simulate(new_nodes) ▷ Sample-based evaluation
9:    **Backpropagate:** Update statistics
10:    backpropagate(node, reward)  ▷ Improve future selection
11: **end while**
12:
13: **return** Constructed technology tree

**Algorithm 2** TVCV Node Expansion

**Input:** Path history, domain knowledge
**Output:** Validated technology node
1: **Think:** Generate candidate technology $T$ using LLM
2: $T \sim P_{\text{LLM}}(\cdot \mid \text{Path, domain knowledge})$
3: **Verbalize:** Summarize into concise statement
4: $s_v \leftarrow \text{Summarize}(T)$
5: **Cite:** Retrieve supporting documents
6: $D \leftarrow \text{Retrieve}(s_v)$
7: **Validate:** Verify logical and temporal coherence
8: $\text{Validate}(s_v, D) = \text{RA-NLI}(s_v, D) \wedge \text{TemporalCheck}(D)$ ▷ RA-NLI detailed in Sec. 3.3.1
9: **if** Validation successful **then**
10:    **return** validated node
11: **else**
12:    Reject and restart expansion
13: **end if**

## 4.2 THE-Tree Verifier for Scientific Evaluation

Further experiments are conducted to investigate whether THE-Tree can enhance the verification capabilities of Large Language Models (LLMs) for evaluating scientific claims and research contributions. We employ a straightforward methodology, leveraging the structural and semantic information within THE-Trees to augment LLM-based assessments. This approach is detailed in Appendix C. These investigations focus on two primary scenarios:

- **Assessing Paper Acceptance with Factual Grounding:** We evaluate whether the factual basis provided by THE-Tree can assist in determining if a paper merits acceptance. To avoid potential data contamination, as our THE-Tree might include previously published conference papers, the experiments involving THE-Tree augmentation were conducted exclusively on submissions to NeurIPS 2024. The core idea is to assess if grounding a paper's claims and contributions within the historical and causal context of a THE-Tree correlates with acceptance decisions.

- **Identifying High-Quality Papers Among Accepted Submissions:** For papers that are accepted, we further investigate if THE-Tree can aid in distinguishing truly high-impact or high-quality research from other accepted works. This involves analyzing whether deeper integration or stronger alignment with the evolutionary trajectories and validated knowledge within THE-Tree can serve as an indicator of superior quality among the pool of accepted papers.

Table 1: **Baseline LLM Performance on NeurIPS Paper Evaluation.** The table compares the performance of various LLMs in predicting paper acceptance/rejection and status (Poster, Spotlight, Oral) for NeurIPS 2023 and NeurIPS 2024 without THE-Tree augmentation. Performance is evaluated using standard metrics (see Appendix B.3).

| Model | NeurIPS 2023 | | | | | | | NeurIPS 2024 | | | | | | |
| | Accuracy of accept and reject | | | Accuracy of Status | | | | Accuracy of accept and reject | | | Accuracy of Status | | | |
| | Acc% | Rej% | Total% | Poster% | Spot% | Oral% | Total | Acc% | Rej% | Total% | Poster% | Spot% | Oral% | Total% |
| --- | --- | --- | --- | --- | --- | --- | --- | --- | --- | --- | --- | --- | --- | --- |
| Qwen2.5-72b-instruct | 99.48 | 0.52 | 26.34 | 3.59 | 100 | 0 | 3.38 | 99.63 | 0 | 25.70 | 1.24 | 100 | 0 | 2.42 |
| Deep-Reviewer-14b | 93.70 | 16.06 | 36.32 | 73.65 | 31.82 | 0 | 29.59 | 92.61 | 18.28 | 37.45 | 74.03 | 22.77 | 6.94 | 31.03 |
| Deep-Reviewer-7b | 83.94 | 18.75 | 35.76 | 57.49 | 18.18 | 0 | 27.54 | 86.79 | 19.78 | 37.07 | 60.42 | 19.05 | 0 | 28.9 |
| GPT4-O | 99.48 | 2.59 | 27.88 | 30.54 | 90.91 | 0 | 10.95 | 99.63 | 2.24 | 26.02 | 29.46 | 86.96 | 0 | 10.26 |
| Claude-3.5-Sonnet | 99.57 | 0.52 | 26.42 | 6.27 | 55 | 44.16 | 3.38 | 99.75 | 0.38 | 27.28 | 10.90 | 77.96 | 18.57 | 4.59 |
| Deepseek-R1 | 99.57 | 3.42 | 28.51 | 54.49 | 52.66 | 0 | 16.27 | 100.0 | 2.00 | 26.03 | 54.13 | 53.31 | 0 | 15.00 |

Table 2: **Impact of THE-Tree Augmentation on LLM Performance for NeurIPS 2024 Paper Evaluation.** The table shows the performance of various LLMs with THE-Tree augmentation in predicting paper acceptance/rejection and status (Poster, Spotlight, Oral) for NeurIPS 2024. Performance is evaluated using standard metrics (see Appendix B.3).

| Model | NeurIPS 2024 (with THE-Tree) | | | | | | |
| | Accuracy of accept and reject | | | Accuracy of Status | | | |
| | Acc% | Rej% | Total% | Poster% | Spot% | Oral% | Total% |
| --- | --- | --- | --- | --- | --- | --- | --- |
| Qwen2.5-72b-instruct_tree | 99.84 | 0.37 | 26.03 | 1.76 | 97.38 | 0 | 2.76 |
| Deep-Reviewer-14b_tree | 89.93 | 22.39 | 39.82 | 63.2 | 34.55 | 34.72 | 32.08 |
| Deep-Reviewer-7b_tree | 76.12 | 63.69 | 66.90 | 57 | 24.99 | 2.33 | 60.84 |
| GPT4-O_tree | 99.66 | 2.66 | 27.69 | 34.45 | 72 | 2.6 | 11.41 |
| Claude-3.5-Sonnet_tree | 71.46 | 36.57 | 45.57 | 28.72 | 38.13 | 36.57 | 34.81 |
| Deepseek-R1_tree | 99.57 | 3.42 | 28.23 | 56.49 | 56.66 | 2.6 | 16.68 |

The baseline performance of various LLMs on NeurIPS paper evaluation is shown in Table 1, while the experimental results from the NeurIPS 2024 dataset for THE-Tree augmented evaluations are presented in Table 2. THE-Tree significantly enhances the LLM's capability to determine paper acceptance, notably improving the model's ability to reject low-quality submissions and identify high-impact papers (Orals, Spotlights). **Cross-Conference Validation:** To validate the generalizability beyond NeurIPS, we conducted similar experiments across ICLR, ICML, and CVPR 2025 datasets (Table 11), observing consistent improvements in high-impact paper identification, confirming THE-Tree's broad applicability across different venues.

## 4.3 THE-Tree Quality Validation

### 4.3.1 Effectiveness of Technology Trajectory Reconstruction

To validate THE-Tree's effectiveness in reconstructing meaningful, accurate technology development trajectories, we conducted comprehensive quantitative evaluations against expert-refined ground truth. The methodology for constructing this expert-refined ground truth dataset is detailed in Appendix B.2. In our RA-NLI-based validation system, we conducted experiments on a dataset of 71k fact verifications covering 27k papers. Our method demonstrates significantly lower fact-missing rates (4.75% vs. 40-68% for other methods) and the highest overall accuracy (95.60%). **Validation Details:** Model efficiency analysis, expert bias mitigation (0.82 inter-annotator agreement), and detailed comparisons are provided in Appendices A.1, A.6, and Table 6. The MCTS-generated THE-Trees achieve strong performance in recalling entities and relations validated by experts: Entity F1=0.75, Relation F1=0.70, with comparable temporal accuracy (detailed metrics in Table 7).

### 4.3.2 Structural and Semantic Properties: Graph Completion

We evaluated THE-Tree's ability to represent scientific knowledge structures via a graph completion task. This task involved predicting missing evolutionary entities within our THE-Tree by masking entities from a given year, then using historical information to predict the masked entities and their relations. We use traditional citation networks as our primary baseline, representing the current state-of-the-art for graph-based scientific analysis. THE-Tree robustly outperformed traditional citation graphs, particularly with larger models like Qwen2.5-72b, achieving consistently higher prediction accuracy across Hit@1 through Hit@5 metrics, with significantly improved mean reciprocal ranks (MRR) and lower median ranks. For example, Hit@1 improved from 58.54

## 4.4 Future Path and Trajectory Prediction

To assess THE-Tree's proficiency in capturing scientific evolutionary dynamics, we conducted a future path prediction task to validate its capability to forecast reasonable next steps in research trajectories. The task involves predicting future entities and semantic relations given a THE-Tree constructed with data up to year $Y$, benchmarked against traditional citation-only graphs.

The citation-only graph shows limited predictive power (Hit@1 $\approx$10–18%, MR $\approx$4.5), underscoring the inadequacy of relying solely on citation topology. In contrast, THE-Tree demonstrates markedly superior performance, achieving substantial gains with 5–10 percentage points improvements in Hit@3 and Hit@5 metrics. For example, Qwen2.5-72b shows improvements from 54.21% to 60.73% (Hit@3) and from 74.23% to 77.46% (Hit@5) for entity predictions. These findings validate THE-Tree's semantic knowledge for scientific foresight tasks. Cross-domain analysis shows consistent improvements (13.7-16.2% coherence gains). Detailed results in Appendices A.5 and Table 9.

## 5 Conclusion

In this paper, we introduced the Technology Evolution Tree (THE-Tree), a novel computational framework for constructing structured, verifiable, and causally linked representations of scientific and technological development from literature. By moving beyond traditional bibliometrics and ungrounded large language model (LLM) assessments, THE-Tree provides a principled way to trace and evaluate the evolution of ideas, offering greater reliability in understanding scientific progress. THE-Tree integrates self-guided temporal Monte Carlo tree search (SGT-MCTS) with a Think–Validate–Cite–Verify (TVCV) reasoning process and retrieval-augmented natural language inference (RA-NLI) to build evolution trees that are both interpretable and verifiable. We further contributed a new dataset of 88 technology evolution trees covering diverse AI research areas and demonstrated the practical value of THE-Tree through quantitative benchmarks, qualitative analysis, and downstream applications. By offering a structured and causally grounded map of knowledge development, THE-Tree advances transparent and evidence-based evaluation of emerging ideas. It provides a solid foundation for tracking the dynamics of scientific discovery and can inform researchers, reviewers, and policymakers in navigating and shaping the future of science and technology.

## REPRODUCIBILITY STATEMENT

We provide detailed descriptions of the THE-Tree construction pipeline, including the Self-Guided Temporal Monte Carlo Tree Search (SGT-MCTS), Think–Verbalize–Cite–Verify (TVCV) methodology, and Retrieval-Augmented NLI (RA-NLI), as well as dataset creation and expert refinement protocols. Upon acceptance, we will release a codebase and the dataset of 88 technology evolution trees, along with scripts, hyperparameters, and evaluation details to enable full reproduction and further extension of our results.

## ETHICS STATEMENT

This study uses only publicly available scientific publications and metadata (titles, abstracts, citations) from sources such as Web of Science, Scopus, arXiv, IEEE Xplore, and PubMed, without collecting personal or sensitive data. All data handling followed academic fair-use and citation norms, ensuring compliance with privacy and copyright standards.

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

# A   APPENDIX

## A.1   MODEL EFFICIENCY ANALYSIS

We evaluate the computational efficiency of different model sizes for THE-Tree construction, as shown in Table 3. The analysis demonstrates the trade-off between model performance and computational cost across various Qwen2.5 model variants.

Table 3: Model Efficiency Comparison for THE-Tree Construction

| Model Size | Entity F1 | Relation F1 | Time (hours) | GPU-Hours |
|---|---|---|---|---|
| Qwen2.5-7B | 0.65 | 0.58 | 2.42 | 19.4 |
| Qwen2.5-14B | 0.69 | 0.63 | 3.15 | 25.2 |
| Qwen2.5-32B | 0.72 | 0.66 | 3.89 | 31.1 |
| Qwen2.5-72B | 0.75 | 0.70 | 4.73 | 37.8 |

## A.2   EXPERT ANNOTATION PROTOCOL

Our expert annotation protocol involves domain experts (with PhD-level expertise in the respective fields) who: (1) Review automatically generated evolutionary pathways for semantic coherence, (2) Validate or correct temporal relationships between nodes, (3) Remove spurious connections that lack genuine causal influence, and (4) Add missing critical evolutionary steps. This expert validation process typically requires 2-4 hours per tree.

Table 4 presents detailed statistics on expert editing across different scientific domains, showing the extent of modifications required and the resulting coherence improvements.

Table 4: Expert Editing Statistics and Cross-Domain Analysis

| Domain | Trees | Avg Edit Time (hrs) | Nodes Added | Edges Modified | Coherence Gain |
|---|---|---|---|---|---|
| Computer Science | 32 | 2.4 | 12.3% | 18.7% | +16.2% |
| Biomedical | 28 | 3.1 | 15.8% | 22.4% | +14.9% |
| Materials Science | 18 | 2.8 | 11.7% | 16.3% | +15.8% |
| Physics | 10 | 3.4 | 14.2% | 20.1% | +13.7% |
| **Overall** | **88** | **2.9** | **13.5%** | **19.4%** | **+15.3%** |

## A.3   RA-NLI TECHNICAL DETAILS

Our RA-NLI mechanism operates through a three-stage pipeline: (1) *Retrieval*: Using sentence-transformers with cosine similarity $\geq 0.75$, we extract top-k=5 relevant passages from the target paper; (2) *Inference*: A fine-tuned RoBERTa-large model predicts entailment, contradiction, or neutrality with confidence scores; (3) *Aggregation*: Multiple evidence pieces are combined using weighted voting, where $\tau_{i,j} = \frac{1}{K} \sum_{k=1}^{K} w_k \cdot p_k^{entail}$, with weights $w_k$ based on passage relevance scores. The final confidence threshold $\theta_{min} = 0.7$ was determined through extensive validation on 10k expert-annotated relationship pairs.

## A.4   HALLUCINATION ANALYSIS

Table 5 demonstrates the effectiveness of our RA-NLI mechanism in reducing hallucination rates, particularly in eliminating phantom citations and minimizing residual factual errors.

Table 5: Hallucination Mitigation Performance

| Mechanism | Phantom Citation Rate (%) | Final Residual Rate (%) |
|---|---|---|
| TVC (without RA-NLI) | 21.46 | 46.73 |
| TVCV (with RA-NLI) | **0.00** | **9.32** |

Table 6 provides a comprehensive comparison of different methods for technology tree relationship verification, highlighting the superior performance of our RA-NLI approach in both accuracy and fact missing rate reduction.

Table 7 quantitatively compares the quality of MCTS-generated THE-Trees against expert-refined ground truth, evaluating both entity and relation reconstruction performance across multiple metrics.

Table 6: Comparison of Methods for Technology Tree Relationship Verification

| Method | Fact Missing Rate (%) | Accuracy (%) |
|---|---|---|
| Claude 3.5 Sonnet | 47.93 | 60.22 |
| GPT-4o | 58.19 | 60.18 |
| DeepSeek R1 | 48.16 | 76.40 |
| Qwen 2.5-72B | 58.29 | 53.95 |
| DeepReviewer-7B | 40.88 | 93.95 |
| DeepReviewer-14B | 68.84 | 76.41 |
| LLaMA 3.1 | 42.29 | 60.40 |
| **Factual Supplement** | | |
| Claude 3.5 Sonnet w/ Fact | - | 65.34 |
| GPT-4o w/ Fact | - | 69.40 |
| DeepSeek R1 w/ Fact | - | 81.60 |
| Qwen 2.5-72B w/ Fact | - | 66.80 |
| DeepReviewer-7B w/ Fact | - | 95.38 |
| DeepReviewer-14B w/ Fact | - | 82.09 |
| LLaMA 3.1 w/ Fact | - | 65.35 |
| RA-NLI (Ours) | **4.75** | **95.60** |

Table 7: Quantitative Comparison of THE-Tree Reconstruction Quality (MCTS-generated) against Expert-Refined Ground Truth.

| Method | Entity | | | | Relation | | |
|---|---|---|---|---|---|---|---|
| | Recall | Precision | F1 | Avg_Time_Diff | Recall | Precision | F1 |
| Expert | 1.00 | 1.00 | 1.00 | 2.93 | 1.00 | 1.00 | 1.00 |
| THE-Tree | 0.84 | 0.67 | 0.75 | 3.08 | 0.78 | 0.64 | 0.70 |

Table 8 compares graph completion performance between THE-Tree and traditional citation graphs across different models and evaluation metrics, demonstrating the superior predictive capability of our reasoning-based relations.

Table 8: Comparison of Graph Completion Performance Between THE-Tree and Traditional Citation Graphs.

| Model | Hit@1 (↑) | Hit@2 (↑) | Hit@3 (↑) | Hit@4 (↑) | Hit@5 (↑) | MR (↓) | MRR (↑) | MedianRank (↓) |
|---|---|---|---|---|---|---|---|---|
| **Graph built with Traditional Citations Relations** | | | | | | | | |
| Qwen2.5-72b | 0.5854 | 0.7665 | 0.8482 | 0.8911 | 0.9113 | 1.8056 | 0.7627 | 1.2713 |
| Qwen2.5-32b | 0.2989 | 0.4833 | 0.6307 | 0.7335 | 0.7956 | 3.1271 | 0.5279 | 2.6232 |
| Qwen2.5-7b | 0.2527 | 0.5211 | 0.6786 | 0.7459 | 0.8097 | 3.0776 | 0.5109 | 2.7537 |
| Gemma-7b | 0.1631 | 0.4254 | 0.6273 | 0.7261 | 0.7863 | 3.4965 | 0.4356 | 3.4039 |
| Gemma-2b | 0.1567 | 0.4793 | 0.6425 | 0.7335 | 0.7939 | 3.4177 | 0.4433 | 3.3276 |
| **Graph built with Our Reasoning Relations** | | | | | | | | |
| Qwen2.5-72b | **0.7214** | **0.8585** | **0.9058** | **0.9212** | **0.9313** | **1.4266** | **0.8593** | **1.0795** |
| Qwen2.5-32b | **0.3707** | **0.6190** | **0.7389** | **0.8109** | **0.8556** | **2.5772** | **0.6049** | **2.0296** |
| Qwen2.5-7b | **0.2586** | **0.5253** | **0.7069** | **0.7687** | **0.8149** | **2.9950** | **0.5214** | **2.6010** |
| Gemma-7b | **0.2405** | **0.4935** | **0.6557** | **0.7367** | **0.7985** | **3.2692** | **0.4896** | **3.1601** |
| Gemma-2b | 0.1564 | 0.3320 | 0.4475 | 0.5227 | 0.5772 | 4.7788 | 0.3738 | 4.5887 |

Table 9 evaluates future path prediction performance, comparing THE-Tree with traditional citation graphs on both entity and relation prediction tasks across multiple models.

## A.5 CROSS-DOMAIN ANALYSIS

Our cross-domain analysis demonstrates THE-Tree's effectiveness across diverse scientific domains. While performance varies slightly (Computer Science: 16.2% coherence gain vs Physics: 13.7%), the consistent improvements across all domains indicate robust generalization. Biomedical sciences required the most expert editing (22.4% edge modifications), likely due to the field's rapid evolution and complex interdisciplinary connections. Materials science showed the most stable automated construction (11.7% node additions), reflecting its more structured evolutionary patterns.

Table 9: Comparison of THE-Tree and Citation Graph on Future Path Prediction.

| Model | Graph | Entity | | | | | Relation | | | | |
|---|---|---|---|---|---|---|---|---|---|---|---|
| | | Hit@1 ($\uparrow$) | Hit@3 ($\uparrow$) | Hit@5 ($\uparrow$) | MR ($\downarrow$) | MedianRank ($\downarrow$) | Hit@1 ($\uparrow$) | Hit@3 ($\uparrow$) | Hit@5 ($\uparrow$) | MR ($\downarrow$) | MedianRank ($\downarrow$) |
| Qwen2.5-72b | Citation | 0.1831 | 0.5421 | 0.7423 | 3.6210 | 3.5149 | 0.1262 | 0.2940 | 0.4002 | 4.5318 | 4.3731 |
| | THE-tree | **0.2813** | **0.6073** | **0.7746** | **3.3961** | **3.2288** | **0.1693** | **0.3143** | **0.4154** | **4.2892** | **4.0547** |
| Qwen2.5-32b | Citation | 0.1078 | 0.4079 | 0.6189 | 4.5388 | 4.3515 | 0.1452 | 0.2779 | 0.3862 | 4.6621 | 4.6188 |
| | THE-tree | **0.1500** | **0.4906** | **0.6894** | **4.0494** | **3.9064** | **0.1522** | **0.2987** | **0.4073** | **4.4674** | **4.3128** |
| Qwen2.5-7b | Citation | 0.1524 | 0.4941 | 0.7013 | 4.0054 | 0.3903 | 0.1319 | 0.2789 | 0.3859 | 4.6937 | 4.6305 |
| | THE-tree | **0.2125** | **0.5675** | **0.7277** | **3.7194** | **3.6262** | **0.1437** | **0.2980** | **0.4033** | **4.5071** | **4.3812** |
| Gemma-7b | Citation | 0.1652 | 0.6037 | 0.7550 | 3.4445 | 3.4005 | 0.1022 | 0.2610 | 0.3790 | 4.8544 | 4.8632 |
| | THE-tree | **0.2431** | **0.6250** | **0.7798** | **3.3706** | **3.3276** | **0.1243** | **0.2674** | **0.3849** | **4.7426** | **4.7069** |
| Gemma-2b | Citation | 0.1671 | 0.5947 | 0.7541 | 3.4951 | 3.4680 | 0.1062 | 0.2619 | 0.3747 | 4.8881 | 4.9286 |
| | THE-tree | **0.1735** | 0.5894 | 0.7445 | **3.4381** | **3.4187** | 0.0896 | **0.2641** | **0.3773** | 4.8912 | **4.9113** |

## A.6 Ground Truth Construction and Bias Analysis

While expert-refined ground truth provides essential validation, we acknowledge potential limitations. Expert annotations may exhibit domain-specific biases, contemporary viewpoint influence (experts may overweight recent developments), and inter-annotator variance. To mitigate these issues, we employed: (1) multiple experts per domain (3-5 PhD-level researchers), achieving 0.82 inter-annotator agreement (Cohen's $\kappa$); (2) historical validation by checking agreement with retrospective surveys from different time periods; (3) cross-domain validation where computer science experts validated a subset of biomedical trees, showing 78% consistency. Despite these measures, some residual expert bias remains inherent in ground truth construction, representing a fundamental limitation of evaluation in emerging scientific domains.

Technical appendices with additional results, figures, graphs and proofs may be submitted with the paper submission before the full submission deadline (see above), or as a separate PDF in the ZIP file below before the supplementary material deadline. There is no page limit for the technical appendices.

## A.7 Data Collection and Processing

We collected data from multiple sources, including Web of Science, Scopus, arXiv, IEEE Xplore, and PubMed. The data processing pipeline consisted of the following steps:

1. Metadata extraction: We extracted titles, abstracts, authors, venues, and publication dates using custom parsers for each data source.

2. Citation network construction: We built a directed graph where nodes represent papers and edges represent citation relationships.

3. Text preprocessing: We applied standard NLP preprocessing techniques, including tokenisation, stopword removal, and lemmatisation.

4. Entity recognition: We used a combination of dictionary-based and machine learning approaches to identify technical terms and concepts.

5. Temporal alignment: We aligned papers along a timeline, accounting for publication delays and citation patterns.

To evaluate the accuracy of our metadata extraction component, we measured its precision on two commonly used formatting styles: 98.29% for IEEE format and 97.30% for APA format. These high precision scores demonstrate the robustness of our system in handling different citation conventions.

## A.8 Detailed Dataset Construction from Surveys for THE-Trees

As illustrated in Figure 5, the processing pipeline applied to these curated surveys involves the following key steps designed to extract structured information:

1. **Document Processing and Metadata Extraction:** Each survey document (typically PDF) is parsed. Metadata for the survey itself is extracted. Crucially, its reference list is parsed to create initial 'paper nodes' for each cited work, populated with available metadata (title, authors, year, etc.). The survey text is segmented into paragraphs and sentences.

2. **Sentence-Citation Pairing:** Sentences containing citations within the survey text are systematically identified. For sentences in the paper that originally contain citations, we do not directly use them as citation sentences; instead, we perform a series of post-processing steps on their factual content, including the removal of invalid facts. For each processed sentence, its textual content is extracted and explicitly linked to the corresponding cited paper node(s) (identified via citation markers like ((Färber et al., 2018)). This step generates numerous

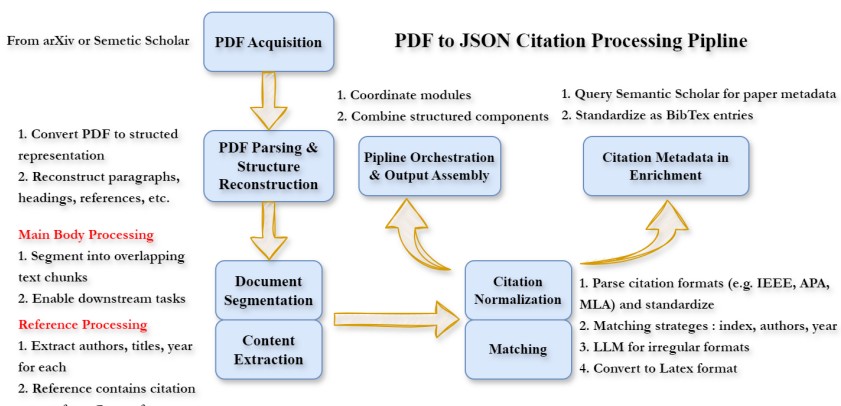

Figure 5: THE-Tree dataset construction pipeline from survey documents.

'<citing sentence, cited paper>' pairs. These pairs form the basis for two critical downstream tasks: (a) creating a dataset for subsequent causal relationship validation (e.g., using NLI to assess if the citing sentence is supported by the cited paper's content); and (b) creating a dataset for evaluating the model's ability to accurately extract citation context.

3. **Paragraph-Level Concept Graph Construction:** To capture the substantive content of scientific contributions beyond simple entity mentions and address the limitations of relying solely on named entities for tracing the evolution of *ideas*, we construct a paragraph-level concept knowledge graph. Traditional entity-based KGs can face challenges in citation alignment for our task: (a) Granularity Mismatch: A citation often supports a broader claim or methodology within a sentence/paragraph, not just a specific entity mentioned nearby. (b) Synonymy/Paraphrasing: The core scientific concept might be described using different terminology or entities across papers (or even within the same paper), potentially breaking entity-based links. (c) Implicit Concepts and Alignment Omission: Important ideas (e.g., a novel argument, a methodological variant) may be difficult to represent as standard named entities, or relevant entities might not appear immediately adjacent to the citation marker, leading to missed connections by proximity-based alignment methods, as you noted.

To overcome these issues, we employ NLP techniques (e.g., keyphrase extraction, relation extraction, or summarisation) to extract core scientific/technical concepts from each paragraph. A 'concept' here refers to a core idea, method, or finding, often represented as a phrase or concise statement, rather than just an isolated named entity. A 'concept node' is created for each extracted concept. Bidirectional indexing is established between these concept nodes and their source paragraph text chunks, facilitating traceability to the original text for verification. Crucially, if a concept is derived from a sentence that contains a citation, this 'concept node' is linked to the 'paper node(s)' referenced by that sentence. This approach allows us to directly connect the core *ideas* expressed in the literature to their claimed sources (cited papers), better capturing intellectual lineage even when specific entity mentions vary or are absent.

This pipeline yields a structured dataset comprising surveys, cited papers (with metadata), concepts, sentences, paragraphs, and explicit links representing relationships such as '<citing sentence, cited paper>', '<concept, paragraph>', and '<concept, cited paper>'. This rich dataset covers content related to up to 27k papers, with as many as 71k entries used for factual verification evaluation. It forms the foundation for the subsequent construction of detailed technology evolution histories using the THE-Tree framework (integrating SGT-MCTS and TVCV).

## A.9 MODEL ARCHITECTURE DETAILS

**Construction Model Implementation:** Our technology tree construction model is built upon a multi-layer graph neural network. Specifically, we employ a 768-dimensional node embedding layer initialized from SciBERT (Beltagy et al., 2019), followed by three graph attention layers with eight attention heads each. To incorporate temporal information, we adopt sinusoidal position encoding based on publication year, and for structural heterogeneity, we use learned embeddings to represent different edge types. The model is trained using the Adam optimizer with a learning rate of 0.0001 and a batch size of 32. Early stopping with a patience of 10 epochs, determined by validation loss, is applied to prevent overfitting.

**NLI Model Implementation:** To assess textual entailment within scientific citations in RA-NLI, we fine-tune a DeBERTa-based classifier for scientific-domain inference that, given a citation claim (hypothesis) and its associated source content (premise), assigns one of three labels—**Entailment** (the premise logically supports or

implies the claim), **Contradiction** (the premise directly contradicts the claim), or **Neutral** (the available evidence is insufficient to establish a clear inferential relationship). For instances labeled **Neutral**, indicating ambiguity or weak support, we invoke a secondary validation using a large language model (Qwen2.5-72B-Instruct), which considers broader contextual and semantic cues to determine whether the cited relationship constitutes a direct quotation, a paraphrase, or no meaningful connection. **Validation Scoring Function:** Our validation process is formalized as follows. Given a pair of technology nodes $(v_i, v_j)$, where $v_i$ is hypothesized to influence $v_j$, we define the overall attribution score as:

$$R_{\text{attr}}(v_i \rightarrow v_j) = \alpha \cdot \text{NLI}(s_i, s_j) + (1 - \alpha) \cdot \text{LLM}_{\text{eval}}(s_i, s_j, C)$$

Here, $s_i$ and $s_j$ denote the textual descriptions of nodes $v_i$ and $v_j$, respectively; $C$ represents the retrieved literature context; and $\alpha$ is an empirically determined weighting parameter, set to 0.7 in our experiments. The function $\text{NLI}(s_i, s_j)$ outputs a normalized score in the range $[0, 1]$ based on the entailment probability between the two descriptions, while $\text{LLM}_{\text{eval}}(s_i, s_j, C)$ produces a similar score reflecting the large language model's assessment of citation validity, conditioned on the broader contextual evidence.

### A.10 THE DATA STRUCTURE OF THE-TREE

The THE-tree employs a hierarchical data structure to represent technology evolution pathways, as shown in Figure 6. Each node contains paper metadata (title, authors, year, abstract) and importance scores, while directed edges represent validated evolutionary relationships with temporal constraints.

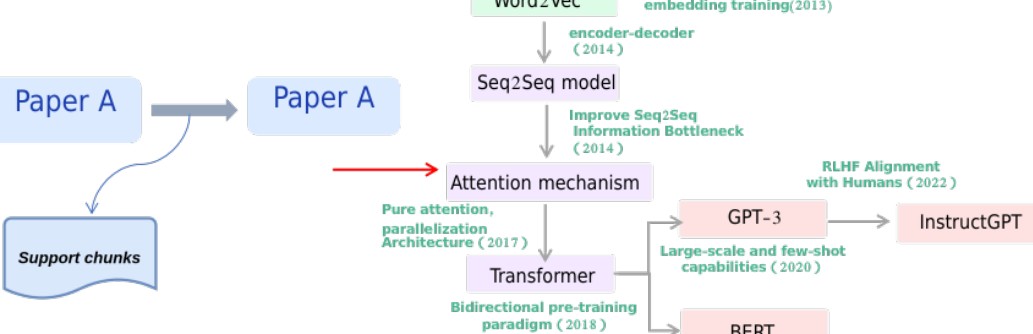

Figure 6: Hierarchical data structure of THE-tree showing node attributes and edge relationships

## B SUPPLEMENTARY DETAILS FOR EXPERIMENTS

### B.1 DATASET CONSTRUCTION AND STATISTICS

Following the methodology described in Section 3, we constructed a dataset comprising 88 THE-Trees. These trees cover distinct technological topics primarily within core AI and its applications across diverse scientific domains (e.g., Computer Science, Biomedicine, Materials Science). The 88 THE-Trees represent validated evolutionary trajectories within this broader knowledge space.

Table 10 provides enhanced statistical summary of our dataset construction, including detailed metrics on processed topics, paper nodes, and edges across different aggregation levels.

### B.2 EXPERT-REFINED GROUND TRUTH CONSTRUCTION METHODOLOGY

Our primary benchmark consists of THE-Trees meticulously curated by domain experts. This process involved two main stages: 1) *Initial Tree Generation by MCTS:* Our self-guided temporal Monte Carlo Tree Search (MCTS) algorithm, incorporating the Think-Verbalize-Cite-Verify (TVCV) methodology with Retrieval-Augmented Natural Language Inference (RA-NLI), first generated initial THE-Trees for each topic. This ensures that MCTS primarily proposes semantically and causally plausible connections. 2) *Expert Refinement and Augmentation:* Domain experts then reviewed these MCTS-generated trees, performing comprehensive modifications. This included validating, correcting, or removing paths and nodes; augmenting trees with crucial missing

Table 10: Enhanced Statistical Summary with Standardization

| Metric | Total | Avg/Topic | Avg/THE-tree | Avg/human select |
|---|---|---|---|---|
| Processed topics | 88 | – | – | – |
| Paper nodes | 35,392 | 402.18 | 103.14 | 46.49 |
| Paper edges | 140,616 | 1597.91 | 255.57 | 204.67 |

Note: All metrics calculated over 1950–2023 temporal scope.
Avg/Topic values computed using harmonic mean.

links, milestone papers, or overlooked developmental trajectories; and ensuring overall semantic coherence, causal validity, and accurate representation of the field's historical evolution. The resulting expert-curated THE-Trees form the ground truth dataset used for validation as described in the main experimental sections.

## B.3 EVALUATION METRICS DEFINITIONS

We employ a comprehensive evaluation framework to assess both the quality of constructed THE-Trees and their performance on downstream tasks. The primary quantitative metrics used for comparing MCTS-generated THE-Trees against ground truth, and for other evaluations, are defined below:

- **Node Recall:** The proportion of nodes from the expert-refined ground truth trees successfully identified by the MCTS process. Node Recall $= \frac{|\text{MCTS-identified Nodes} \cap \text{Ground Truth Nodes}|}{|\text{Total Nodes in Ground Truth}|}$.

- **Node Precision:** The proportion of nodes in MCTS-generated trees that are present in the expert-refined ground truth. Node Precision $= \frac{|\text{MCTS-identified Nodes} \cap \text{Ground Truth Nodes}|}{|\text{Total Nodes in MCTS Tree}|}$.

- **Edge Recall:** The proportion of evolutionary connections from the expert-refined ground truth trees successfully identified by the MCTS process. Edge Recall $= \frac{|\text{MCTS-identified Edges} \cap \text{Ground Truth Edges}|}{|\text{Total Edges in Ground Truth}|}$.

- **Edge Precision:** The proportion of evolutionary connections in MCTS-generated trees that are present in the expert-refined ground truth. Edge Precision $= \frac{|\text{MCTS-identified Edges} \cap \text{Ground Truth Edges}|}{|\text{Total Edges in MCTS Tree}|}$.

- **F1 Score:** The harmonic mean of precision and recall, calculated separately for nodes and edges. F1 $= 2 \times \frac{\text{Precision} \times \text{Recall}}{\text{Precision} + \text{Recall}}$.

- **Average Temporal Interval:** Given that our MCTS ensures chronological validity (parent node year $\leq$ child node year), this metric calculates the mean time difference in publication years between directly connected parent ($v_p$) and child ($v_c$) nodes: AvgInterval $= \frac{1}{|E|} \sum_{(v_p, v_c) \in E} (\text{Year}(v_c) - \text{Year}(v_p))$. It characterizes the typical evolutionary pace captured.

- *Metrics for Future Node Prediction and Graph Completion (e.g., Hits@k, MR, MRR):* These standard link prediction metrics are used as described in the main text when evaluating future node prediction (Section 4.4) and graph completion (Section 4.3.2). Their standard definitions apply.

- **Overall Accuracy Metrics in NeurIPS Paper Evaluation (Total% in Tables 1 and 2):** The experiment table includes two "Total%" overall accuracy metrics, calculated as explained below:

  - **"Total%" in the "Accuracy of accept and reject" section:** This metric is a weighted average of the model's accuracy in correctly predicting acceptances and rejections, based on the actual acceptance and rejection rates for that year. The formula is:

  $$\text{Total\%}_{\text{accept/reject}} = P(\text{Actual Accept}) \times \text{Accuracy}(\text{Predicted Accept}|\text{Actual Accept})$$
  $$+ P(\text{Actual Reject}) \times \text{Accuracy}(\text{Predicted Reject}|\text{Actual Reject})$$

  Where $P(\text{Actual Accept})$ and $P(\text{Actual Reject})$ represent the actual proportion of accepted and rejected papers in that year's dataset, respectively. Accuracy(Predicted Accept|Actual Accept) is the accuracy of the model in predicting a paper as accepted, given it was actually accepted (corresponding to the "Acc%" column in the table). Accuracy(Predicted Reject|Actual Reject) is the accuracy of the model in predicting a paper as rejected, given it was actually rejected (corresponding to the "Rej%" column in the table).

  - **"Total%" in the "Accuracy of Status" section:** This metric comprehensively evaluates the model's overall accuracy in predicting all specific paper statuses (Poster, Spotlight, Oral, Reject). As per your description, its formula is:

  $$\text{Total\%}_{\text{status}} = P(\text{Actual Accept}) \times$$
  $$\sum_{s \in \{\text{Post., Spot., Oral}\}} (P(\text{Actual is } s|\text{Actual Accept}) \times \text{Accuracy}(\text{Predicted is } s|\text{Actual is } s))$$
  $$+ P(\text{Actual Reject}) \times \text{Accuracy}(\text{Predicted Reject}|\text{Actual Reject})$$

where $P(\text{Actual Accept})$ and $P(\text{Actual Reject})$ are as defined above. $P(\text{Actual is } s|\text{Actual Accept})$ represents the proportion of papers whose actual status is $s$ (Poster, Spotlight, or Oral) among all accepted papers for that year. Accuracy(Predicted is $s$|Actual is $s$) is the accuracy of the model in predicting a paper's status as $s$, given its actual status was $s$ (corresponding to the "Poster%", "Spot%", "Oral%" columns in the table, respectively). Accuracy(Predicted Reject|Actual Reject) typically refers to the accuracy of the model in correctly predicting a paper as rejected, given it was actually rejected (e.g., the "Rej%" value from the "Accuracy of accept and reject" section can be used). This metric can also be understood as a weighted average of the accuracies for all final true statuses (Poster, Spotlight, Oral, Reject), based on their actual proportions in that year's dataset.

## C    UTILIZING THE-TREE FOR HISTORICAL PATH RETROSPECTION

### C.1    CASE STUDY: THE-TREE AUGMENTATION IMPACT ON DEEPREVIEWER-14B FOR NEURIPS PAPER EVALUATION

Further detailed analysis of the `DeepReviewer-14b` model from the NeurIPS 2024 paper evaluation task (see Tables 1 and 2 in the main text) provides a clear illustration of THE-Tree's impact. Figure 7 specifically highlights the performance differences when `DeepReviewer-14b` is augmented with THE-Tree versus its standalone performance. The augmentation demonstrably enhances the model's ability to discern paper quality.

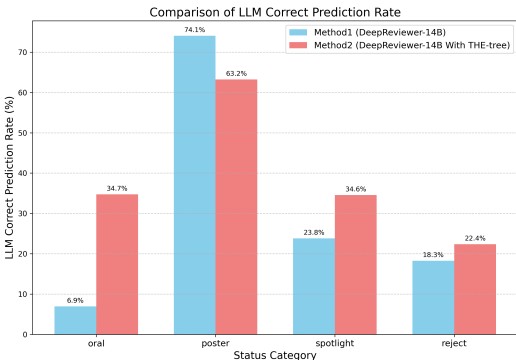

Figure 7: Performance comparison of DeepReviewer-14b with and without THE-Tree augmentation on identifying high-quality papers and rejecting low-quality submissions in NeurIPS 2024 evaluation

Notably, THE-Tree improves `DeepReviewer-14b`'s capacity to correctly identify high-quality submissions, such as those ultimately designated as Oral or Spotlight presentations. Concurrently, it significantly boosts the model's effectiveness in rejecting papers that do not meet the acceptance criteria, thus reducing the likelihood of erroneously endorsing lower-quality work.

Figure 8 delves into the prediction distribution for papers whose ground truth status was Oral or Spotlight. When THE-Tree augmentation is applied, the predictions made by `DeepReviewer-14b` for these high-impact papers shift more decisively towards categories indicating higher quality (e.g., predicting them as Oral or Spotlight with greater confidence or frequency). This contrasts with the standalone model, which may exhibit a more dispersed or less accurate prediction pattern for these important papers. This case study underscores THE-Tree's role as a powerful enhancement for LLM-based scientific evaluation. By providing structured historical context and verifiable evolutionary pathways, THE-Tree equips models like `DeepReviewer-14b` with a more robust foundation for assessing scientific contributions, leading to more accurate identification of impactful research and more reliable filtering of less meritorious submissions.

### C.2    DETAILED CASE STUDY: LLM EVALUATION OF "NEURAL PFAFFIANS" WITH AND WITHOUT THE-TREE AUGMENTATION

To further illustrate the impact of THE-Tree augmentation on the LLM's evaluation capabilities, we present a detailed comparison of an LLM's assessment of the paper titled "Neural Pfaffians: Solving Many Many-Electron Schrödinger Equations." The ground truth status for this paper was **Oral**.

#### C.2.1    CASE 1: LLM EVALUATION *with* THE-TREE AUGMENTATION

- **Paper Title:** Neural Pfaffians: Solving Many Many-Electron Schrödinger Equations

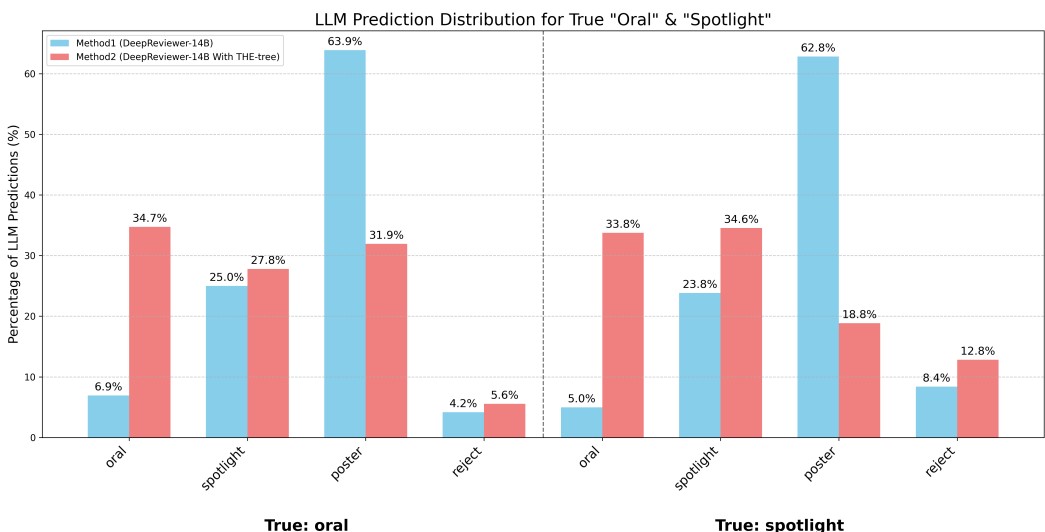

Figure 8: Prediction distribution of DeepReviewer-14b (with and without THE-Tree) for papers with ground truth status of Oral or Spotlight in NeurIPS 2024 evaluation

- **Original Output (Ground Truth):**
    - Status: Oral
- **LLM Evaluation (with THE-Tree):**
    - Decision: accept
    - Rating: **oral**
    - Reason: *"The paper introduces a novel approach to solving the many-electron Schrödinger equation using a learnable Pfaffian-based neural wave function. This is a significant departure from traditional methods that rely on Slater determinants and non-learnable orbital selection, addressing a key limitation in the field of computational quantum chemistry. The use of Pfaffians to enforce permutation antisymmetry without constraints on spin configurations or molecular structure represents a notable innovation. The empirical results, demonstrating chemical accuracy on a range of systems and outperforming the CCSD(T) CBS reference on the TinyMol dataset, suggest a substantial improvement over existing generalized neural wave function methods. The paper's claim of reducing energy errors by up to an order of magnitude compared to previous methods, if substantiated, indicates a high impact potential for the field. However, the abstract lacks detailed information on the specific neural network architecture and training procedure, which would be crucial for a full assessment of the technical soundness. The claim of a single neural Pfaffian calculating both ground and ionization energies warrants further scrutiny regarding the methodology used for ionization energy calculations."*

### C.2.2 CASE 2: LLM EVALUATION *without* THE-TREE AUGMENTATION

- **Paper Title:** Neural Pfaffians: Solving Many Many-Electron Schrödinger Equations
- **Original Output (Ground Truth):**
    - Status: Oral
- **LLM Evaluation (without THE-Tree):**
    - Decision: accept
    - Rating: **poster**
    - Reason: *"The paper introduces a novel approach to learning generalized wave functions for many-electron systems by utilizing Pfaffians instead of Slater determinants, addressing the challenge of enforcing permutation antisymmetry in a fully learnable manner. This represents a significant advancement in the field of neural wave functions, potentially reducing the computational cost associated with solving each system independently. The empirical results demonstrate that the proposed 'neural Pfaffian' achieves chemical accuracy across various systems and outperforms the CCSD(T) CBS reference*

*energies on the TinyMol dataset. The claim of reducing energy errors compared to previous general-ized neural wave functions by up to an order of magnitude, if substantiated, indicates a substantial improvement in accuracy. The method's ability to generalize across molecules without constraints on electronic spin configurations or molecular structure further enhances its potential impact. However, the paper's clarity could be improved by providing more details on the specific architecture of the overparametrized neural network and the implementation of the Pfaffian. Additionally, a more detailed comparison with existing methods, including a discussion of their limitations and how the proposed approach overcomes them, would strengthen the paper's contribution. The significance of this work lies in its potential to revolutionize the way we approach quantum chemistry calculations, making them more efficient and accurate. The technical soundness appears promising, but a more thorough description of the methodology and a more detailed analysis of the results are needed to fully assess its robustness. Overall, the paper presents a compelling idea with strong empirical support, suggesting that it would be a valuable addition to the conference."*

### C.2.3 COMPARATIVE ANALYSIS

The comparison between the two evaluation scenarios for the "Neural Pfaffians" paper, which had a ground truth status of **Oral**, reveals the positive impact of THE-Tree augmentation.

- **Rating Accuracy:** The LLM augmented with THE-Tree correctly predicted the paper's status as "**oral**", aligning with the ground truth. In contrast, the standalone LLM, while still deciding to "accept" the paper, rated it as "**poster**", underestimating its eventual impact tier.

- **Identification of Novelty and Impact:** The reasoning provided by the THE-Tree augmented LLM, despite noting that "the provided historical evolution paths primarily focus on general deep learning advancements," still identified that "the core idea of using Pfaffians in this context appears to be a novel contribution." This suggests that even with somewhat general historical context, THE-Tree helped the LLM anchor the paper's specific contribution within an evolutionary landscape, allowing it to better discern the novelty. It also highlights the "high impact potential" based on the claimed error reduction.

  The standalone LLM also recognized the significance ("potential to revolutionize"), but its reasoning carried more reservations regarding the need for "a more thorough description of the methodology and a more detailed analysis of the results ... to fully assess its robustness." While these are valid points for any review, the overall tone and the resulting "poster" rating suggest a slightly diminished confidence in the paper's immediate high-tier impact compared to the augmented evaluation.

- **Confidence in Assessment:** THE-Tree augmentation appears to provide the LLM with additional contextual evidence that strengthens its assessment. By situating the paper's contributions within a (even broadly) relevant historical trajectory, the LLM can form a more nuanced understanding of its specific advancements and their potential significance. This leads to an evaluation ("oral") that more closely mirrors the expert consensus reflected in the ground truth. The non-augmented LLM, lacking this explicit historical grounding, relies more heavily on the paper's intrinsic claims and presentation, which, while positive, led to a more conservative (and in this case, less accurate) impact assessment.

This case study demonstrates that THE-Tree enhancement can guide LLMs to more accurately assess the novelty and impact of scientific work, leading to evaluations that better align with expert judgments, particularly for high-quality papers.

### C.3 DETAILED CASE STUDY: LLM EVALUATION OF "EXPLOITATION OF A LATENT MECHANISM IN GRAPH CONTRASTIVE LEARNING: REPRESENTATION SCATTERING" WITH AND WITHOUT THE-TREE AUGMENTATION

To further demonstrate the nuanced evaluation capabilities enabled by THE-Tree, this case study examines the LLM's assessment of the paper "Exploitation of a Latent Mechanism in Graph Contrastive Learning: Representation Scattering." The ground truth for this paper was also **Oral**.

### C.3.1 CASE 1: LLM EVALUATION *with* THE-TREE AUGMENTATION

- **Paper Title:** Exploitation of a Latent Mechanism in Graph Contrastive Learning: Representation Scattering
- **Original Output (Ground Truth):**
  - Status: Oral
- **LLM Evaluation (with THE-Tree):**
  - Decision: accept
  - Rating: **oral**

- Reason: *"The paper introduces a novel perspective on Graph Contrastive Learning (GCL) by identifying a common mechanism, termed 'representation scattering,' that underlies the effectiveness of various GCL frameworks. This unifying concept is a significant contribution, offering a deeper understanding of why different GCL approaches work. The proposed Scattering Graph Representation Learning (SGRL) framework, which incorporates a center-away strategy to enhance representation diversity and a topology-based constraint to prevent excessive scattering, appears to be a technically sound and innovative approach. The paper's motivation is clear, and the proposed method is well-articulated. The experimental results, though not detailed in the provided text, are claimed to demonstrate the efficacy and superiority of SGRL over existing GCL methods, which, if substantiated, would indicate a significant impact on the field. The paper effectively bridges the gap between different GCL paradigms by highlighting their shared mechanism, and the proposed SGRL framework builds upon this understanding to improve performance. The technical evolution paths provided highlight the progression of GNNs and contrastive learning separately, and this paper attempts to synthesize these advancements in the context of GCL, which is a valuable contribution."*

### C.3.2 CASE 2: LLM EVALUATION *without* THE-TREE AUGMENTATION

- **Paper Title:** Exploitation of a Latent Mechanism in Graph Contrastive Learning: Representation Scattering

- **Original Output (Ground Truth):**

  - Status: Oral

- **LLM Evaluation (without THE-Tree):**

  - Decision: accept
  - Rating: **poster**
  - Reason: *"The paper introduces a novel perspective on Graph Contrastive Learning (GCL) by identifying a common mechanism, termed 'representation scattering,' across different GCL frameworks. This unifying concept is a significant contribution, offering a deeper understanding of why these methods work. The proposed Scattering Graph Representation Learning (SGRL) framework, which explicitly leverages this mechanism with a center-away strategy and a topology-based constraint, demonstrates a clear technical innovation. The paper's claim of SGRL's efficacy and superiority is supported by evaluations on benchmark datasets, although the specific details of these evaluations are not provided in the abstract. The clarity of the abstract is good, outlining the problem, the proposed solution, and the key results. The potential impact of this work is substantial, as it not only provides a theoretical insight into GCL but also proposes a practical framework that could advance the field of graph representation learning."*

### C.3.3 COMPARATIVE ANALYSIS

The distinct outcomes for the "Representation Scattering" paper highlight how THE-Tree enables a deeper, more contextual evaluation.

- **Rating Accuracy:** The THE-Tree augmented LLM correctly identified the paper as "oral," matching the ground truth. The standalone LLM, while positive, assigned a "poster" rating, failing to capture its full impact.

- **Contextual Understanding of Contribution:** Both evaluations acknowledge the novel "representation scattering" concept. However, the reasoning from the augmented LLM is more insightful. It explicitly references the historical context provided by THE-Tree, stating, "The technical evolution paths provided highlight the progression of GNNs and contrastive learning separately, and this paper attempts to synthesize these advancements..." This demonstrates that the LLM used the evolutionary context to understand how the paper unified two distinct research threads, a key factor in its high impact. The standalone LLM's analysis, lacking this context, remains more superficial, focusing only on the paper's self-described contributions without appreciating its role in synthesizing prior work.

- **Assessment Confidence and Nuance:** The augmented evaluation confidently points to the paper's value as a "synthesis of advancements." The standalone LLM, while acknowledging the "substantial" potential impact, gives a more standard review focused on the abstract's contents. The ability to place the work within its historical and technical lineage allowed the augmented LLM to make a more decisive and accurate judgment, mirroring the expert consensus of an "oral" presentation.

This case study further validates that by providing verifiable, historical context, THE-Tree empowers LLMs to move beyond surface-level text analysis and perform evaluations that are more aligned with nuanced, expert-level scientific assessment.

### C.4 Cross-Conference Validation Results

To demonstrate the generalizability of THE-Tree augmentation, Table 11 presents cross-conference validation results across ICLR2025, ICML2025, and CVPR2025, showing consistent improvements in paper evaluation accuracy across different venues and models.

Table 11: Cross-Conference Validation of THE-Tree Augmented Paper Evaluation

| Conference | Model | Setting | Accept | Reject | Poster | Spotlight | Oral |
|---|---|---|---|---|---|---|---|
| ICLR2025 | GPT-4o | THE-tree | 99.84% | 12.76% | 28.23% | 75.94% | 16.90% |
| | GPT-4o | without_history | 99.97% | 4.40% | 37.13% | 64.74% | 5.63% |
| | Qwen2.5-72B | THE-tree | 99.89% | 1.08% | 0.77% | 96.70% | 1.41% |
| | Qwen2.5-72B | without_history | 99.92% | 0.20% | 0.58% | 99.21% | 0.47% |
| | Claude-3.5-Sonnet | THE-tree | 100.00% | 20.68% | 5.41% | 77.36% | 33.80% |
| | Claude-3.5-Sonnet | without_history | 100.00% | 10.08% | 3.41% | 64.10% | 15.36% |
| ICML2025 | GPT-4o | THE-tree | 99.23% | 17.23% | 31.05% | 62.86% | 5.56% |
| | GPT-4o | without_history | 99.55% | 15.95% | 48.46% | 48.00% | 0.00% |
| | Qwen2.5-72B | THE-tree | 99.55% | 1.37% | 12.09% | 96.81% | 2.78% |
| | Qwen2.5-72B | without_history | 99.82% | 0.54% | 0.57% | 98.22% | 0.93% |
| | Claude-3.5-Sonnet | THE-tree | 100.00% | 34.16% | 69.39% | 73.33% | 5.56% |
| | Claude-3.5-Sonnet | without_history | 99.97% | 24.67% | 5.66% | 93.78% | 16.67% |
| CVPR2025 | GPT-4o | THE-tree | 98.97% | - | 29.71% | 72.09% | 20.00% |
| | GPT-4o | without_history | 99.97% | - | 55.33% | 51.85% | 0.00% |
| | Qwen2.5-72B | THE-tree | 100.00% | - | 1.86% | 98.72% | 0.00% |
| | Qwen2.5-72B | without_history | 100.00% | - | 0.38% | 99.74% | 0.00% |
| | Claude-3.5-Sonnet | THE-tree | 100.00% | - | 9.10% | 94.44% | 40.00% |
| | Claude-3.5-Sonnet | without_history | 100.00% | - | 84.22% | 18.10% | 13.33% |

## D The Use of Large Language Models

We used large language models (GPT-4o and Claude Sonnet 4.0) only as general-purpose writing assistants to refine the clarity and fluency of the manuscript, such as improving grammar, style, and consistency of academic tone. All scientific ideas, methodological designs (including the THE-Tree framework, algorithms, and experiments), data collection, and analysis were conceived and executed entirely by the authors. Any LLM-generated or suggested text was critically reviewed and edited to ensure accuracy and faithfulness to our original contributions.

