# OpenReview forum: "THE-Tree: Can Tracing Historical Evolution Enhance Scientific Verification and Reasoning?"
_ICLR.cc/2026/Conference — ICLR 2026 Conference Withdrawn Submission_

### Official Review · Reviewer_MHaW · 2025-10-29

**Soundness:** 2
**Presentation:** 2
**Contribution:** 2
**Rating:** 4
**Confidence:** 3

**Summary:**

This paper introduces THE-Tree (Technology History Evolution Tree), a computational framework for constructing structured, causally-linked representations of scientific evolution from literature. The authors employ Self-Guided Temporal Monte Carlo Tree Search (SGT-MCTS) with a Think-Verbalize-Cite-Verify (TVCV) methodology to build evolution trees where nodes represent papers and edges represent validated evolutionary relationships. The framework uses Retrieval-Augmented Natural Language Inference (RA-NLI) to verify relationships. The authors construct 88 THE-Trees across AI domains and demonstrate improvements in graph completion (8-14%), future development prediction (~10%), and paper evaluation tasks.

**Strengths:**

### **1. Originality**

**1.1 Novel problem formulation and framework design.** The paper addresses a genuine bottleneck in AI-driven scientific discovery—validating AI-generated hypotheses—through an original lens of constructing structured, causally-linked historical evolution trees. The TVCV (Think-Verbalize-Cite-Verify) methodology represents a creative combination of Monte Carlo Tree Search, LLM generation, and retrieval-augmented validation that hasn't been previously explored for scientific knowledge graph construction. The explicit focus on causal evolutionary relationships rather than simple citation links is a meaningful departure from traditional bibliometric approaches.

**1.2 Innovative validation mechanism.** The RA-NLI (Retrieval-Augmented Natural Language Inference) component addresses a critical weakness in LLM-based systems—hallucination—by grounding proposed relationships in actual textual evidence from papers. The reported reduction of phantom citations from 21.46% to 0% represents a significant technical contribution to ensuring factual accuracy in automated knowledge graph construction.

---

### **2. Quality**

**2.1 Substantial dataset contribution.** The construction of 88 THE-Trees covering 27k papers with 71k verified relationships represents a significant effort. The dataset spans diverse AI research areas and includes expert refinement with reasonable inter-annotator agreement (0.82 Cohen's κ), providing a valuable resource for the research community.

**2.2 Comprehensive multi-task evaluation.** The paper evaluates THE-Tree across multiple downstream tasks (graph completion, future prediction, paper evaluation) rather than a single application, demonstrating breadth of potential impact. The cross-conference validation (NeurIPS, ICLR, ICML, CVPR) in Table 11 strengthens generalizability claims within the AI domain.

**2.3 Rigorous construction pipeline.** The SGT-MCTS algorithm with temporal constraints and the multi-stage validation process (NLI model + LLM fallback + expert refinement) shows methodological sophistication. The explicit handling of temporal consistency and the composite reward function balancing node importance, path coherence, and link validity demonstrate careful system design.

---

### **3. Clarity**

**3.1 Well-structured presentation.** The paper provides clear motivation, systematic methodology description, and extensive experimental results. Figure 1 effectively communicates the high-level approach and contrasts with existing methods. The formal edge definition (Section 3.1) with explicit temporal and validation constraints provides mathematical precision.

**3.2 Comprehensive appendices.** The extensive supplementary material (Sections A-C) provides implementation details, model architectures, dataset statistics, and case studies that support reproducibility. The worked examples in Appendix C (Neural Pfaffians, Graph Contrastive Learning) help readers understand practical application.

---

### **4. Significance**

**4.1 Addresses timely problem.** As AI systems increasingly generate scientific hypotheses, the need for automated validation becomes critical. The paper tackles this important challenge with a structured approach that could influence how the research community thinks about AI-assisted scientific evaluation.

**4.2 Practical improvements demonstrated.** The reported improvements—8-14% in graph completion, ~10% in future prediction, and substantial gains in paper evaluation—suggest potential real-world utility. The case studies show concrete examples where THE-Tree augmentation improved paper quality assessment.

**4.3 Foundation for future research.** By releasing the dataset and framework, the paper enables follow-up work on scientific knowledge representation, automated reviewing, and computational science history.

**Weaknesses:**

### **W1. Circular Reasoning in Benchmark Construction and Evaluation (Critical)**

**Issue:** The benchmark is constructed using LLMs (TVCV with LLM proposals, RA-NLI with LLM fallback for ambiguous cases) and expert refinement of LLM outputs, then used to demonstrate that LLM-based evaluation improves when augmented with this benchmark. This circularity undermines claims about THE-Tree capturing "authentic patterns" of scientific evolution versus capturing what LLMs believe evolution looks like.

**Specific concerns:**

- Expert refinement starts from LLM proposals (anchoring bias) rather than independent construction
- Ground truth for graph completion/future prediction is "papers retrospectively included in THE-Tree," not independently validated milestones
- Improvements may reflect LLMs recognizing their own construction patterns rather than genuine scientific understanding

**Requested evidence:**

**W1.1.** Ablation where experts build THE-Trees independently without seeing LLM outputs, measuring agreement between expert-first vs. LLM-first construction

**W1.2.** Validation against independent benchmarks (e.g., test-of-time award papers, highly-cited papers 5 years post-publication)

**W1.3.** Analysis of whether different LLMs produce consistent THE-Trees for the same domain (inter-model reliability)

**Suggested fix:** Section 4 should include independent validation experiments that don't rely on LLM-generated or LLM-refined ground truth.

------

### **W2. Relationship Type Assignment: Incomplete Documentation (Critical)**

**Issue:** The paper introduces three semantic relationship types {causal, enabling, foundational} as a core advantage over citation networks but never explains how these types are assigned, validated, or distinguished.

**Missing information:**

- No operational definitions differentiating the three types with concrete criteria
- No description of whether types are LLM-generated, expert-assigned, or algorithmically determined
- No inter-annotator agreement specifically for relationship type classification
- No confusion matrix or error analysis for type assignments
- No justification for why three types are sufficient versus more granular taxonomies (e.g., "extends," "refutes," "optimizes," "applies-to-new-domain")

**Impact on claims:** Without clear type assignment methodology, the claimed semantic superiority over citation networks cannot be properly evaluated. If types are LLM-generated without validation, this represents a significant quality gap.

**Requested addition:** Add Section 3.3.2 explicitly describing:

**W2.1.** How relationship types are determined at each stage (LLM generation, RA-NLI, expert refinement)

**W2.2.** Clear definitions with examples distinguishing each type

**W2.3.** Inter-annotator agreement for type classification

**W2.4.** Discussion of type taxonomy design choices


------

### **W3. Unfair Experimental Comparisons (Major)**

**Issue:** The comparisons between THE-Tree and traditional citation networks are uncontrolled for information content, making it impossible to isolate THE-Tree's specific structural contribution.

**Specific problems:**

*Graph Completion (Table 8):*

- THE-Tree: typed edges, importance scores, confidence scores, retrieved evidence passages
- Citation network: binary edges only
- This is comparing information-rich vs. information-sparse representations
- Better performance may simply reflect having more features, not superior causal structure

*Paper Evaluation (Tables 1-2):*

- LLM + THE-Tree: receives historical context, related papers, evolutionary paths
- LLM alone: only title + abstract
- Any form of historical context (RAG retrieval, survey paragraphs, citation network paths) might provide similar improvements

**Missing baselines:**

**W3.1.** Citation networks augmented with paper importance scores (from PageRank, citation counts, or Semantic Scholar metrics)

**W3.2.** LLM + RAG retrieval of related papers (matching THE-Tree's information content)

**W3.3.** LLM + relevant survey paragraphs about the topic

**W3.4.** Citation networks with citation context text (available from tools like Semantic Scholar)

**Requested experiments:** Table comparing:

- THE-Tree (full)
- THE-Tree with random edge types (ablation)
- Citation network + importance scores
- Citation network + citation contexts
- RAG baseline with equivalent information

This would isolate whether improvements stem from THE-Tree's causal structure versus simply providing more context.

------

### **W4. Temporal Leakage in Future Prediction (Major)**

**Issue:** The "future prediction" experiment (Section 4.4, Table 9) claims to predict developments after year Y using data up to Y, but the LLMs used were trained on data beyond Y, creating temporal leakage.

**Specific problem:**

- THE-Tree built "up to 2019" to predict 2020-2023
- But Qwen2.5-72b, GPT-4, Claude-3.5-Sonnet were all trained on data including 2020-2023
- These models already "know" what came after 2019 from their training
- This isn't true prediction but retrofitting/recognition

**Additional concerns:**

- Very low baseline (citation network Hit@1: 18.31%) suggests task setup issues
- Ground truth is papers "retrospectively included in THE-Tree," not independently validated breakthroughs
- No discussion of this limitation in the paper

**Required clarifications:**

**W4.1.** State the training data cutoffs for all LLMs used (Qwen2.5-72b, Gemma-7b, etc.)

**W4.2.** If training data includes the prediction period, acknowledge this as a limitation

**W4.3.** Redesign experiment using models with training cutoffs before prediction period, or

**W4.4.** Design truly prospective evaluation: make predictions, wait for future developments, evaluate against actual outcomes

**Alternative formulation:** Frame as "retrospective pattern matching" rather than "future prediction" and acknowledge the limitation.

**Questions:**

# Questions and Suggestions for Authors

## **A. Semantic Relationship Types (Critical Clarification Needed)**

### **A1. How are relationship types assigned?**

The paper states edges have types {causal, enabling, foundational} but never explains how these are determined.

**Questions:**

- Is the relationship type generated by the LLM during the TVCV "Think" step?
- Do experts validate or modify relationship types during refinement?
- What are the operational definitions distinguishing "causal" vs "enabling" vs "foundational"?
- What is the inter-annotator agreement specifically for relationship type assignments?

**Why this matters:** Without clear type assignment methodology, the claimed semantic advantage over citation networks cannot be evaluated.

**Suggestion:** Add a subsection explicitly describing the relationship typing process with examples and validation metrics.

------

### **A2. Are three types sufficient?**

**Questions:**

- How was the three-type taxonomy derived? Why not more granular (e.g., "extends," "refutes," "optimizes," "applies-to-new-domain")?
- Did experts request additional relationship types during refinement?
- Can you provide confusion matrix or disagreement analysis for the three types?

**Why this matters:** If types frequently overlap or are ambiguous, the semantic structure may not provide meaningful improvement over simpler representations.

------

## **B. Circular Reasoning and Validation Independence (Major Concern)**

### **B1. How do you address potential circular reasoning?**

**The concern:**

```
THE-Tree built by: LLM proposals → RA-NLI validation (NLI + LLM) → Expert refinement of LLM outputs
THE-Tree used for: Improving LLM-based evaluation
```

**Questions:**

- Have you measured whether improvements simply reflect LLMs "recognizing their own construction patterns"?
- Can you provide ablation where experts build THE-Trees independently (without seeing LLM outputs) and compare agreement?
- What percentage of expert refinements are additions vs. modifications vs. deletions of LLM proposals?

**Why this matters:** If THE-Tree primarily captures "what LLMs think evolution looks like" rather than "actual evolution," its value for LLM evaluation is questionable.

**Suggestion:** Conduct independent expert construction experiment and report agreement metrics.

------

### **B2. RA-NLI validation - how robust is it?**

**Questions:**

- What NLI model is used? Is it fine-tuned on scientific text or general domain?
- For the LLM fallback (when NLI score is 0.6-0.8), what percentage of edges require this?
- Can a cited paper "acknowledge contribution" without representing intellectual lineage (e.g., "Like [AlexNet], we use CNNs" vs. actually building on AlexNet's ideas)?
- Can you provide false positive/negative analysis of RA-NLI with expert evaluation?

**Why this matters:** If RA-NLI has high false positive rate (accepting spurious relationships), the validation isn't actually filtering noise.

**Suggestion:** Report precision/recall/F1 of RA-NLI against expert judgments on a held-out validation set of relationship proposals.

------

## **C. Experimental Design and Fair Comparisons**

### **C1. Graph completion - is the comparison fair?**

**The issue:** THE-Tree has typed edges, importance scores, confidence scores, evidence passages. Citation networks have binary edges.

**Questions:**

- Can you compare against citation networks augmented with similar metadata (e.g., citation context, paper importance scores from Semantic Scholar)?
- What happens if you give the model THE-Tree structure but random edge types - does performance degrade?
- Is the improvement from THE-Tree's specific causal structure, or just from having richer information?

**Why this matters:** Comparing information-rich vs. information-sparse representations doesn't isolate the value of your specific approach.

**Suggestion:** Add ablation studies:

- THE-Tree with random edge types
- Citation network + paper importance scores
- Citation network + citation context text

------

### **C2. Future prediction - how do you avoid temporal leakage?**

**The concern:** You build THE-Tree "up to year Y" but use LLMs trained on data beyond Y, which already "know" what came after Y.

**Questions:**

- What is the training data cutoff of the LLMs used (Qwen2.5-72b, etc.)?
- If LLM training includes 2020-2023 data, how can it fairly "predict" 2020-2023 developments?
- Can you re-run experiments using LLMs with training cutoffs before the prediction period?
- What constitutes "correct" future prediction - papers added to THE-Tree retrospectively, or papers that actually had impact?

**Why this matters:** If models already know the future, this isn't prediction but retrofitting.

**Suggestion:**

1. Clearly state LLM training cutoffs
2. Design truly prospective evaluation (make predictions, wait for future, evaluate)
3. Or use historical holdout with appropriate model cutoffs

------

### **C3. Paper evaluation - what's the fair baseline?**

**The issue:** LLM+THE-Tree vs. LLM alone isn't isolating THE-Tree's value - any historical context might help.

**Questions:**

- How does performance compare to:
  - LLM + citation network paths (same number of papers)?
  - LLM + RAG retrieval of related papers?
  - LLM + relevant survey paragraphs?
- The baseline GPT-4o achieves 0% Oral identification for NeurIPS 2024 - why is this so low? Is the prompt or task setup problematic?
- Can you show the actual prompts used for with/without THE-Tree conditions?

**Why this matters:** Need to establish whether THE-Tree's structure provides unique value vs. any form of historical context.

**Suggestion:** Add comparison conditions isolating THE-Tree's contribution:

- Same information content, different structures
- Ablate specific THE-Tree components (types, confidence scores, evidence)

------

## **E. Generalization and Scope**

### **E1. Paradigm shifts and revolutionary science**

You mention handling paradigm shifts (Section 3.1) but provide no empirical validation.

**Questions:**

- Can you show examples of THE-Tree successfully capturing a paradigm shift?
- How does the framework handle papers that fundamentally break from prior trajectories?
- Does the survey-based initialization bias against revolutionary work not yet covered in surveys?

**Suggestion:** Demonstrate on historical paradigm shift (e.g., attention mechanisms replacing RNNs, AlexNet reviving deep learning).

------

## **H. Clarifications on Methodology**

### **H1. Expert refinement protocol**

**Questions:**

- How many experts per THE-Tree? What are their qualifications?
- How much time did experts spend per tree (you say 2-4 hours, but what's the distribution)?
- What specific instructions were experts given?
- For the 15.3% coherence improvement - how is "coherence" measured objectively?

------

### **H2. Node importance scores**

**Questions:**

- You define S(v) = γ·Sgraph(v) + (1-γ)·SLLM(v) - how is γ chosen?
- Is γ constant across domains or tuned per THE-Tree?
- How sensitive are results to γ values?

------

### **Summary: Most Critical Questions for Rebuttal**

1. **How do you address circular reasoning** (LLMs building benchmarks to evaluate LLMs)?
2. **How are relationship types assigned** and what's the validation?
3. **Can you provide fair comparisons** (THE-Tree vs. citation networks with equal information)?
4. **How do you avoid temporal leakage** in future prediction experiments?
5. **Can you demonstrate value beyond AI domains** with non-AI THE-Trees?

These responses would significantly impact my assessment of the paper's contributions and validity.

---

---

### Official Review · Reviewer_CMH3 · 2025-10-31

**Soundness:** 3
**Presentation:** 3
**Contribution:** 2
**Rating:** 6
**Confidence:** 4

**Summary:**

This paper introduces THE-Tree (Technology History Evolution Tree), a computational framework for constructing structured, verifiable representations of scientific evolution from literature. The approach combines Self-Guided Temporal Monte Carlo Tree Search (SGT-MCTS) with a novel Think-Verbalize-Cite-Verify (TVCV) methodology and Retrieval-Augmented Natural Language Inference (RA-NLI) to build domain-specific evolution trees that capture causal relationships between scientific papers. The authors construct 88 THE-Trees across diverse domains and demonstrate improvements in graph completion, future prediction, and paper evaluation tasks.

**Strengths:**

1. The paper is generally well-written and organized, with clear motivation and comprehensive experiments. The main concepts are explained well, and the figures effectively illustrate the framework.
2. The challenge of evaluating AI-generated scientific ideas is timely and important, especially given the proliferation of LLM-based research tools. The demonstrated improvements in paper evaluation tasks (especially for identifying high-impact papers) have clear practical applications for peer review and research assessment.
3. The RA-NLI mechanism effectively reduces hallucination rates (from 21.46% to 0% for phantom citations), addressing a critical issue in LLM-based approaches.
4. The paper includes extensive experiments across multiple tasks, datasets, and venues.

**Weaknesses:**

1. The approach requires existing survey papers as starting points, which may not be available for emerging or niche research areas. This dependency could limit applicability.
2. The ground truth seems to rely on expert annotation with potential biases. While acknowledged, the paper doesn't provide sufficient mitigation strategies beyond inter-annotator agreement.
3. Missing ablation studies on key components like:
    - The impact of different weighting schemes in the reward function
    - The contribution of individual components in TVCV
    - The effect of different confidence thresholds
4. The paper doesn't adequately discuss when THE-Tree might fail or produce incorrect evolutionary paths, particularly for paradigm shifts or interdisciplinary work.
5. While the paper evaluation results are impressive, the broader claim of enabling "AI-driven scientific discovery" is not sufficiently validated. The experiments primarily focus on paper assessment rather than hypothesis generation or discovery.

Minor: I have noticed that the computational cost (4.73 hours per tree on 8×A100 GPUs) is high, which could limit accessibility.

**Questions:**

1. How does THE-Tree approach handle papers that contribute to multiple research threads or interdisciplinary work that doesn't fit cleanly into a single evolutionary path?
2. How would THE-Tree perform in rapidly evolving fields where survey papers quickly become outdated?
3. What is the sensitivity of the results to the choice of threshold parameters (e.g., θ_min = 0.7, similarity threshold ≥ 0.75)?
4. Can you provide concrete examples of cases where THE-Tree failed to capture important evolutionary relationships?
5. How does the approach handle conflicting evolutionary narratives that might exist in contentious research areas? I am very curious about this.

---

### Official Review · Reviewer_Xdgz · 2025-11-01

**Soundness:** 2
**Presentation:** 2
**Contribution:** 2
**Rating:** 2
**Confidence:** 3

**Summary:**

This paper proposes THE-tree, a way of constructing augmented scientific knowledge graphs which helps address the problem of evaluating AI-generated scientific ideas. One issue is that manual verification of these ideas is labour intensive and slow, whereas LLMs may miss related literature when judging novelty of an idea. They propose a process to generate domain-specific evolution trees from the scientific literature as an alternative to traditional citation networks. In the THE-tree framework, each paper is represented by a node containing metadata about that paper, and each node is associated with an importance score (weighted combination of its structural significance within the citation graph and semantic relevance as assessed by an LLM). Edges between papers represent different types of relationships, but importantly they represent significant contributions or influence of one paper on another rather than just one citing another. They construct THE-trees for 88 topics and assess the constructed trees with graph completion tasks as compared to baseline knowledge graphs, as well as performance in identifying accepted papers and oral/spotlight papers from Neurips 2024.

**Strengths:**

**S1**: I think the problem that this paper is targeting is very timely and relevant as there is an increasing interest in using LLMs for scientific ideation, and LM-generated papers are being increasingly submitted to conferences. The construction of THE-trees or similar approaches could help LMs evaluate whether proposed ideas (whether generated by LMs or humans) are novel and help with LM-as-a-judge approaches.

**S2**: The tree construction method is detailed in great depth, and I think the ideas used to construct the trees generally make sense. Even if not used in their exact form, I think these ideas could potentially inform future designs of scientific citation networks.

**Weaknesses:**

Unfortunately, I think that there are some significant issues in the experimental setup of this paper, and once fixed this paper could be a much stronger contribution. That's why I presently recommend rejection, but I would like to say that I think the paper has potential.

**W1**: The paper is framed as a way to improve scientific idea evaluation, but the contribution of THE-tree seems significantly more narrow from the methods. Of the experiments, the most convincing ones compare THE-tree to traditional citation networks, whereas the remaining experiment focuses on predicting acceptance/rejection decisions for a conference, which while somewhat correlated with idea quality, is also a proxy for the motivating usage (improving evaluation of scientific ideas).

**W2**: From Table 1 vs Table 2, it seems like the gains from including THE-tree are mixed and model dependent, with some models having a much more modest improvement (e.g. Qwen2.5 72b goes from 25.7 -> 26.03). Additionally, the larger gains on models such as Deepreviewer 7B and Claude Sonnet 3.5 seem to be driven by higher rejection accuracy while the acceptance accuracy actually goes down. This seems to be because models are predicting accept on almost everything by the numbers on the Acc% column? This raises questions about the prompt given to LMs (see questions). I recommend that the prompt be revised and that F1 score or brier score be used instead.

**W3**: Currently, THE-tree has many different components and it is unclear to me which parts of the methodology drive gains in performance over a traditional citation network. It would be helpful to show ablations for the different modules such as the RA-NLI edges, node importance, TVCV and MCTS.

**W4**: The details of the traditional citation network compared with are not clear to me. In addition to adding more details, I think the authors should also compare with something like semantic scholar’s “highly influential citations” as that also aims to isolate more meaningful citations. (Valenzuela et al 2015)

**Questions:**

- What was the prompt given to models for the accept/reject and paper status (oral/spotlight/poster) classification? It seems like in Table 1 and 2 the LLMs are voting to accept almost everything

- Which traditional citation networks did you compare with and are there any details about them or how they were constructed?

---

### Official Review · Reviewer_7fD9 · 2025-11-01

**Soundness:** 2
**Presentation:** 2
**Contribution:** 2
**Rating:** 4
**Confidence:** 4

**Summary:**

This paper introduces the THE-Tree, a computational framework  that aims to make scientific reasoning and verification more evidence-based by reconstructing causally linked, temporal evolution trees of scientific ideas from literature.

The motivation stems from two issues: (i) LLMs hallucinate and lack awareness of historical contexts, and (ii) Citation networks lack explicit causal and logical coherence.

THE-Tree addresses these by representing each paper as a node with metadata and an importance score. Connecting papers via edges and using the Retrieval Augmented NLI mechanism to validate that an edge has textual, factual support.

Overall, the paper argues that structured, causal historical modeling can ground AI-driven scientific discovery.

**Strengths:**

1. __Ambitious Vision:__ The problem tackles an important bottleneck in AI4Science: how to verify machine-generated hypotheses.
2. __Methodological Coherence:__ Integrates Monte-Carlo search, LLM Reasoning, and NLI-based factual grounding in a coherent pipeline.
3. __Dataset:__ One of the largest structured attempts to model scientific evolution.

**Weaknesses:**

1. __Potential Data Leakage:__ Experiments evaluate LLMs on past conference papers while using training data drawn from broad scientific corpora. Given that many papers already appear on preprint servers months before review, there is a leak of data leakage --- THE-Tree or its LLM components may have already seen these texts. The authors do not report any leakage check.
2. __Ground Truth Causality:__ "Causal" relations are defined linguistically (NLI entailment) rather than via experimental or citation-intent evidence. Hence, the framework captures semantic relatedness, not necessarily genuine causal influence. The validation by experts is small and does not demonstrate that the resulting graph encodes true intellectual lineage rather than sophisticated co-citation semantics.
3. __Miscellaneous:__
   - Metrics like "Hit@k" for graph completion are borrowed from link-preduction and not fully aligned with the notion of scientific evolution.
   - Constructing one tree requires approximately 5 hrs on 8xA100 GPUs, which is heavy. The authors ignored the maintenance cost, as domains may evolve monthly.
   - Phrases like "computable science history" are sometimes overstating claims; at its core, this is a structured bibliometric + LLM system.

**Questions:**

1. How were causal labels validated by experts? Was disagreement quantified beyond the reported kappa?
2. What happens if we remove RA-NLI or use plain MCTS without LLM guidance?
3. Can incremental updates be done - eg, adding 2026 papers without rebuilding trees from scratch?
4. How does the framework prevent reinforcing historical biases or stifling paradigm shifts?

---

### Note · Authors · 2026-01-26

I have read and agree with the venue's withdrawal policy on behalf of myself and my co-authors.

---

### Meta-Review · Area_Chair_Swom · 2025-12-08

**Summary:**

The paper has 3 borderlines scores leaning towards rejection along with one review (Xdgz) that votes clearly for rejection.

In particular, the reviewers identify many strengths including novel and ambitious problem formulation, clear presentation and the presentation of a new dataset.

However there are concerns shared amongst the reviewers. In particular, that there may be data leakage (Reviewers 7fD9, MHaW), issues on methodology and metric used for comparison (Reviewers 7fD9, MHaW), missing ablation experiments (Reviewers Xdgz, CMH3), and that it falls short of its broader aim of AI for scientific discovery (Reviewers Xdgz, CMH3).

As such, even though the paper makes interesting contribution, in its current state it is not ready. I advise the authors to carefully consider the reviewer concerns. Specifically on data leakage and evaluation methodology for future revisions.

**Reviewer Concerns:**

The authors did not respond to the reviewers. Hence all concerns of the reviewers are currently unaddressed.

**Reviewer Scores:**

Again since the authors did not respond. The reviewers are unlikely to have changed their score even if the discussion continued.

---

### Decision · Program_Chairs · 2026-01-26

Reject